# Genome-wide screening in pluripotent cells identifies *Mtf1* as a suppressor of mutant huntingtin toxicity

Giorgia Maria Ferlazzo [1,8,9], Anna Maria Gambetta [1,2,9], Sonia Amato [2,3,9], Noemi Cannizzaro [1], Silvia Angiolillo[1], Mattia Arboit [1], Linda Diamante [2], Elena Carbognin [2], Patrizia Romani [1], Federico La Torre [2], Elena Galimberti[4], Florian Pflug [4], Mirko Luoni [5], Serena Giannelli [5], Giuseppe Pepe[6], Luca Capocci[6], Alba Di Pardo[6], Paola Vanzani [1], Lucio Zennaro [1], Vania Broccoli [5,7], Martin Leeb [4], Enrico Moro [1], Vittorio Maglione [6] & Graziano Martello [2] ✉

Huntington's disease (HD) is a neurodegenerative disorder caused by CAG-repeat expansions in the huntingtin (*HTT*) gene. The resulting mutant HTT (mHTT) protein induces toxicity and cell death via multiple mechanisms and no effective therapy is available. Here, we employ a genome-wide screening in pluripotent mouse embryonic stem cells (ESCs) to identify suppressors of mHTT toxicity. Among the identified suppressors, linked to HD-associated processes, we focus on Metal response element binding transcription factor 1 (*Mtf1*). Forced expression of *Mtf1* counteracts cell death and oxidative stress caused by mHTT in mouse ESCs and in human neuronal precursor cells. In zebrafish, *Mtf1* reduces malformations and apoptosis induced by mHTT. In R6/2 mice, *Mtf1* ablates motor defects and reduces mHTT aggregates and oxidative stress. Our screening strategy enables a quick in vitro identification of promising suppressor genes and their validation in vivo, and it can be applied to other monogenic diseases.

HD is the most widespread monogenic neurodegenerative disorder among the Caucasian population (prevalence of ~7-11 individuals out of 100,000 people)[1,2]. Due to its autosomal dominant inheritance, a single copy of mutate *HTT* gene is sufficient to confer pathological phenotypes, both in patients and in experimental models[3].

The disease is caused by an abnormal expansion (>36) of a CAG triplet in *HTT*, resulting in the formation of a mHTT protein, containing polyglutamine (polyQ) repeats[4]. The wild-type (WT) HTT includes from 9 to 35 Q residues at the $NH_2$ terminus and was implicated in

neural tube formation, resistance to apoptotic stimuli, transcriptional control of brain-derived neurotrophic factor and related genes[5-11]. Indeed, the polyQ repeats expansion confers a toxic gain-of-function to mHTT, leading to abnormal accumulation of aggregation-prone proteins, increased sensitivity to glutamate toxicity, mitochondrial damage and misregulation of the transcriptional program[12-15]. However, it is still hard to know which processes are early causative events and which are consequences. Although HTT protein is ubiquitously expressed[16], HD is characterised by cell-population specific damages[14],

[1]Department of Molecular Medicine, Medical School, University of Padua, 35131 Padua, Italy. [2]Department of Biology, University of Padova, Via U. Bassi 58B, 35131 Padua, Italy. [3]Department of Neuroscience, University of Padova, Via Belzoni, 160, 35131 Padua, Italy. [4]Max Perutz Laboratories Vienna, University of Vienna, Vienna Biocenter, Dr Bohr Gasse 9, 1030 Vienna, Austria. [5]Division of Neuroscience, San Raffaele Scientific Institute, 20132 Milan, Italy. [6]IRCCS Neuromed, 86077 Pozzilli, Italy. [7]CNR Institute of Neuroscience, 20854 Vedrano al Lambro, Italy. [8]Present address: Aptuit (Verona) S.r.l., an Evotec Company, Campus Levi-Montalcini, 37135 Verona, Italy. [9]These authors contributed equally: Giorgia Maria Ferlazzo, Anna Maria Gambetta, Sonia Amato. ✉e-mail: graziano.martello@unipd.it

loss of efferent medium spiny neurons in the striatum of the basal ganglia[17] and massive degeneration of cortical structures[18].

Despite significant advances on HD pathogenesis, no effective therapies are available. Targeting some of the cellular processes impaired in HD gave promising results in animal models[19–23], nonetheless, all clinical trials to date have failed[24]. For this reason, we did not focus a priori on a specific cellular process. We chose instead to screen unbiasedly, on a genome-wide scale, for factors able to reduce mHTT toxicity. To this aim, we used mouse ESCs, which were already used to study the deregulation of transcription[25] and metabolism[26] due to mHTT. Mouse ESCs bear an intact genome that is highly amenable to modification by piggyBac (PB) mediated-insertional mutagenesis, allowing the generation of large-scale mutant libraries successfully used for genetic screenings[27].

Our unbiased and functional approach allowed us to identify more than 100 genes as potential suppressors of mHTT toxicity. After extensive in vitro characterisation, *Mtf1* appeared as a potent suppressor of mHTT-induced cellular toxicity. In human neural precursor cells (NPCs), MTF1 rescued cell death and oxidative stress caused by mHTT. Similarly in a zebrafish HD model[28], *Mtf1* expression alleviated the morphological defects and cell death caused by mHTT. In R6/2 murine HD model[29], *Mtf1* was delivered via brain-penetrating adeno-associated virus (AAV)-vector and a strong amelioration of the motor defects was observed, together with a reduction in mHTT aggregate formation. Collectively, our results indicate that *Mtf1* expression suppressed the detrimental effects caused by mHTT in vitro and in two in vivo HD models, indicating that delivery of *Mtf1* might represent a therapeutic strategy against HD.

## Results

### Establishment and characterisation of mHTT-expressing mouse ESCs

We generated mouse ESCs stably expressing an N-terminal fragment of either mutant (128 CAG repeats) or WT (15 CAG repeats) *HTT*, named Q15 and Q128 cells, respectively (Fig. 1a). Expression of *mHTT* did not cause loss of pluripotency markers (Supplementary Fig. 1a). Translation of mutant and WT form of Q128 and Q15 HTT soluble proteins in cells was confirmed by Western Blot (Fig. 1b). Despite comparable mRNA levels (Supplementary Fig. 1a, left), Q128 HTT protein signal was lower than Q15 HTT (Fig. 1b), in agreement with previous studies on knock-in mouse ESCs and immortalised striatal cells[25,30–35], indicating differences in translation or detectability of mHTT. Aggregates of mHTT were not detected (Supplementary Fig. 1b), in line with previous studies in pluripotent stem cells[25,34,36]. We then asked whether the expression of Q128 HTT would induce cell toxicity. To this aim, we measured the number of cells obtained over 4 days and observed a significant decrease in Q128 cell viability as compared to Q15 cells (Fig. 1c). Conversely, Q15 cells expanded robustly. Similar results were obtained with an independent parental mouse ESC line (Supplementary Fig. 1c-e). Moreover, Q128 cells displayed increased cell death (Fig. 1d and Supplementary Fig. 1f) and increased production of Reactive Oxygen Species (ROS) (Fig. 1e and Supplementary Fig. 1g), as previously reported[24,26,37]. Collectively, Q128 HTT expression impaired mouse ESCs viability.

To further characterise our Q128 HTT expressing mouse ESC model, we performed transcriptome analysis and identified differentially expressed genes (DEGs) in Q128 versus Q15 cells (Fig. 1f and Supplementary Data 1). Gene list enrichment analysis (Fig. 1g) identified misregulation of genes associated with processes implicated in HD pathogenesis, such as metabolism of steroids, glycogen and glutathione[38,39], regulation of histone modifications (i.e. methylation and acetylation)[40], protein ubiquitination[41] and metal homoeostasis[42,43,44]. Moreover, among the DEGs we found genes previously identified as HD biomarkers, or found altered in their expression in HD models, such as *Creb3l3*, *Cox7b2*, *Kalrn* and *Tspo*[45–50] (Fig. 1f).

Overall, we conclude that Q128 cells recapitulate molecular features associated with HD.

### A gain-of-function screening for suppressors of mHTT toxicity

With the aim of performing a genetic screening, we looked for experimental conditions that would exacerbate the toxic effects of mHTT, in order to facilitate the isolation of fully resistant mutants. We reasoned that by treating Q128 cells with different cell stressors we could induce their death, allowing the identification of genetic mutants resistant to the toxic effects of mHTT (Fig. 2a). We searched for inhibitors that were previously shown to worsen the effect of mHTT[51–53] acting on biological processes implicated in HD, such as autophagy, the proteasomal degradation system or mitochondrial metabolism[26,41,51]. We then selected a panel of inhibitors and titrated them to find doses that were not lethal in parental mouse ESCs (Supplementary Fig. 2a). Next, we treated Q128 cells with the inhibitors and observed that MG132 and Tamoxifen further reduced viability of Q128 cells (Fig. 2b). Therefore, they were chosen as stressors selectively inducing cell death in Q128 cells to be used for the genetic screening (Fig. 2a). Two independent stressors were exploited, affecting unrelated biological processes, to increase our chances to identify candidate genes acting specifically on mHTT, rather than on the stressors themselves.

For the screening, we chose a gain-of-function approach. This has the advantage that candidates are selected among all genes present in the intact genome of mouse ESCs, not only among genes expressed in mouse ESCs or expressed in response to mHTT, as for loss-of-function approaches, thus widening our potential for discovery. To generate our genetic mutants, we used PB vectors that are based on transposons, DNA elements that stably integrate into the genome[54]. Electroporation of a PB vector in the presence of the transposase leads to random integration into TTAA sites that are abundant in the genome. The PB vector we used (pGG134, Fig. 2a) was optimised for gain-of-function screenings in mouse ESCs[54] and consists of the murine stem cell virus (MSCV) enhancer/promoter followed by a splice donor site, which allows the over-activation of endogenous genes flanking the site of integration. We electroporated Q128 cells with the pGG134 vector or a PB vector encoding for GFP (PB_GFP), as a control. After treatment with exogenous stressors for 5 days, the surviving colonies were counted. Parental mouse ESCs expressing GFP robustly proliferated in the presence of stressors, while very few Q128 cells expressing GFP survived MG132 or Tamoxifen treatment (Fig. 2c). In contrast, Q128 cells electroporated with pGG134 formed significantly more resistant colonies in presence of both stressors (Fig. 2c), indicating the successful generation of mutants resistant to Q128 HTT.

We individually picked and expanded a total of 44 mutant clonal lines that emerged from 4 mutagenesis experiments, named MG# or T#, according to their derivation in presence of the stressors, MG132 or Tamoxifen, respectively. In order to identify which gene was activated in individual clones, we performed Splinkerette-PCR, which allowed amplification of genomic regions flanking the site of integration. For 35 clones, we obtained 1 or 2 major bands corresponding to the genomic regions at the 5' and 3' ends of the PB vector (Fig. 2d). Splinkerette-PCR bands were then excised and sequenced. Sequence alignment to the reference genome allowed the identification of the precise site of integration and orientation in each mutant cell line. For instance, in the MG15 clone the pGG134 was found inserted upstream of the *Kdm5b* gene and in the correct orientation allowing its activation (Fig. 2d, bottom panel). We identified 23 genes flanking pGG134 (complete list of integration sites in Supplementary Fig. 2b) representing genes potentially conferring resistance to mHTT, or suppressors.

The integration of the PB vector in the proximity of a given gene should lead to its overexpression. We verified by qPCR that endogenous *Kdm5b* transcripts appeared upregulated in the MG15 clone compared to Q128 cells electroporated with PB_GFP or to parental

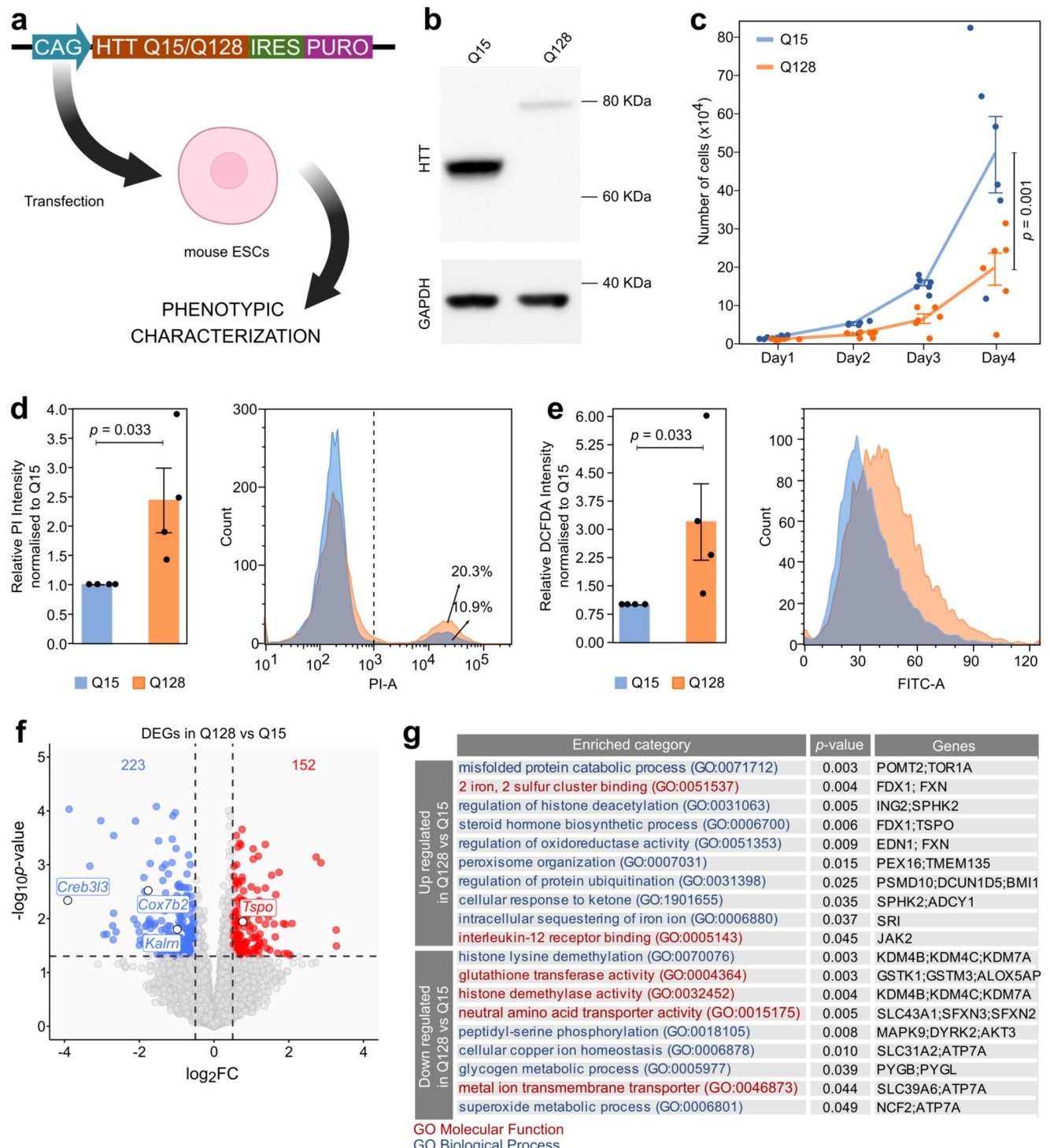

mouse ESCs (Fig. 2e). We characterised 4 additional mutant clones, whose integration sites were unequivocally identified and corresponded to *Fbxo34* (MG21 clone), *Mtf1* (MG18), *Synj2* (MG9) and *Arid1b* (T4). In each clonal line, we verified the upregulation of the cognate genes relative to controls (Fig. 2e).

To validate the ability of the screening procedure to select for mutants resistant to toxicity induced by Q128, we exposed individually the five selected clones to both exogenous stressors. All clones survived comparably to parental mouse ESCs, while Q128 cells electroporated with PB_GFP vector showed reduced survival. Clonal lines were resistant to both stressors (Fig. 2f and Supplementary Fig. 2c-d) suggesting that the mutagenesis procedure led to the activation of

genes conferring resistance to mHTT, rather than resistance to MG132 or Tamoxifen themselves. As the clones might have acquired resistance to mHTT by simply silencing the Q128 *HTT* transgene, we monitored both *HTT* mRNA (Supplementary Fig. 2e) and protein (Fig. 2g), which were robustly detected in all clones. Of note, only for clone MG18, we observed a mild reduction (~17%) in mHTT protein levels. We conclude that the screening procedure led to the identification of 5 genes as bona fide suppressors of toxicity induced by mHTT.

**Network analysis of candidate suppressors of mHTT toxicity**

Prompted by these results, we decided to extend our screening to the whole genome (Fig. 2a). To do so, we collected entire populations of

**Fig. 1 | Establishment and characterisation of mHTT-expressing mouse ESCs.**
**a** Strategy for generation and characterisation of mouse ESCs (Rex1GFP-d2) expressing N-terminal fragment of either mutant (128 CAG repeats) or WT (15 CAG repeats) *HTT* by DNA transfection and puromycin selection, named Q15 and Q128 cells, respectively. Created with BioRender.com. **b** Western Blot for HTT confirmed the correct production of a 80 kDa and a 65 kDa form of HTT protein in Q128 and Q15 cells. GAPDH was used as loading control. The Western Blot shown is representative of 2 independent experiments with 2 technical replicates. **c** Proliferation assay of Q128 and Q15 cells. Lines indicate the mean ± standard error of the mean (SEM) of 6 independent experiments shown as dots. *P*-values were calculated with Two-way Repeated Measure ANOVA. **d** Measurement of cell death by PI uptake and Flow Cytometry. Left: the fraction of PI-positive cells (on the right of the dashed line) was calculated for each sample and fold-changes (FCs) were calculated relative to the Q15 samples. Bars indicate the mean ± SEM of 4 independent experiments shown as dots. Right: representative flow cytometry plots. *P*-values were calculated

with one-tailed one sample Mann-Whitney U test. **e** Left: measurement of H$_2$DCFDA median fluorescence as an evaluation of ROS production in Q15 versus Q128 cells. Bars indicate the mean ± SEM of 4 independent experiments shown as dots. FCs were calculated relative to the Q15 samples. Right: representative flow cytometry profiles. *P*-values were calculated with one-tailed one sample Mann-Whitney U test. **f** Transcriptome analysis of Q128 and Q15 cells. Down-regulated (log$_2$ FC < −0.5 and *p*-value <0.05) and Up-regulated (log$_2$ FC > 0.5 and *p*-value <0.05, indicated by dashed lines) genes are indicated in blue and red, respectively. *P*-values were calculated with two-tailed Wald Test with no adjustment. N = 6 biological replicates for Q15 and n = 7 for Q128 cells. **g** Gene list enrichment analysis of genes down- or up-regulated in Q128 cells, compared to Q15, identified in Fig. 1f. Categories from the Gene Ontology database for Biological processes or molecular functions are shown in blue or red, respectively. *P*-values were calculated by two-tailed Fisher Exact test using Enrichr software.

resistant clones and used Splinkerette-PCR followed by Next-Generation Sequencing (NGS) to map the integration sites in large populations of mutants[55]. We analysed 17 mutant populations obtained from 4 independent experiments. We detected more than 10,000 integration events, corresponding to 804 unique integration sites and, after stringent statistical analysis, we identified 107 genes as candidate suppressors of mHTT cytotoxicity. Integration site distribution analysis (Supplementary Fig. 3a) revealed that pGG134 integrations occurred randomly in the genome with no preference for any given chromosomes. The genes identified in individual clones (Supplementary Fig. 2b), or very closely related members of the same gene family (e.g. *Kdm1b*, *Kdm2b*, *Kdm4c* and *Kdm5b*) were also identified by NGS (Fig. 3a).

Next, we performed a network analysis to obtain insights on how genes identified in mutant clones and by NGS could confer resistance to mHTT. To this aim, we exploited HDNetDB[56], a database that integrates molecular interactions with several manually curated HD-relevant datasets. We obtained a network depicting physical and regulatory interactions between the genes identified in HTT-resistant mutants and HD-related genes (Fig. 3b). The interactome's main nodes correspond to genes identified in mutant clones (purple label) and almost all of them interact with several HD related genes (blue label). Notably, among the HD-related genes, we found several hits from the NGS screening (green label, Fig. 3b), further indicating the consistency and complementarity of the two datasets. This network is particularly enriched in biological processes linked to HD (Supplementary Fig. 3b-c). Moreover, for this network, HDNetDB retrieved several gene sets that integrate molecular interactions with HD and neurodegeneration, as significantly overrepresented (Fig. 3c). Statistically enriched categories were HD genes derived through text mining (HDTM), HD Therapeutic Target Genes (HDTTG) – a curated set of genes that were previously identified as potential therapeutic targets in HD[57], drug-targets, HTT-interacting proteins, Neurological Diseases Gene Association (NDGA) genes that have a genetic association with neurological diseases, as indicated in the Genetic Association Database[58] and NDMOD (Neurodegeneration Modifiers) is derived from independently compiled gene lists comprising genetic modifiers of neurodegeneration identified in various model systems[59]. We concluded that both the genetic screening approaches led to the identification of potential suppressor genes highly associated with HD-related biological processes.

**Secondary validation of mHTT suppressors**
Next, we transfected Q128 cells with a vector containing cDNA of candidate genes under the control of a constitutive promoter, aiming to confirm that our candidates were able to confer resistance to mHTT (Fig. 4a). For such validation experiments, we selected *Mtf1*, *Kdm5b*, and *Fbxo34*, which were identified in mutant clones. We also included *Kdm2b*, identified multiple times in the genome-wide screening (Fig. 3a).

Firstly, we checked whether the expression of *Mtf1*, *Kdm2b*, *Kdm5b* and *Fbxo34* was increased in cells expressing each candidate,

named Q128_Mtf1, Q128_Kdm2b, Q128_Kdm5b and Q128_Fbxo34 cells, respectively. We observed high levels of candidate genes expression in all cell lines generated as compared to controls (Fig. 4b). HTT mRNA levels in Q128 cells were unaltered (Fig. 4c), while HTT protein was slightly reduced (15-22%) by all candidates (Fig. 4d). We checked whether the proliferation of all Q128 cell lines was affected by the co-expression of our candidates. Expression of *Mtf1* and *Kdm2b* led to a significantly increased number of Q128 cells (Fig. 4e-f and Supplementary Fig. 4a). We then exposed all Q128 cell lines to stressors (MG132 or Tamoxifen). Among all candidates, *Mtf1* stood out for its capacity to confer resistance to mHTT in the presence of both stressors (Fig. 4g). We concluded that expression of *Mtf1* consistently grants resistance to mHTT, confirming the phenotype observed in the clone. Therefore, *Mtf1* was chosen for further molecular characterisation and in vivo studies.

MTF1 might exert a generic protection against cell death, rather than a specific protection against mHTT. To investigate the specificity of MTF1 effects, we generated both parental mouse ESCs (E14) and Q15 cells stably expressing *Mtf1* (E14_Mtf1 and Q15_Mtf1, Supplementary Fig. 4b). No differences in the proliferation rate were observed (Fig. 4h and Supplementary Fig. 4c). We exposed Q15_Mtf1 cells to 7 different compounds inducing cell death, including MG132 and Tamoxifen. Notably, MTF1 did not show any protective effect (Supplementary Fig. 4d). We conclude that in mouse ESCs that do not express mHTT, MTF1 does not exert any generic protective effect.

**Mtf1 regulates HD-related processes**
MTF1 is a transcription factor that acts as a sensor for various stress conditions in cells[60]. Upon accumulation of metals (such as Cadmium, Zinc or Iron) but also hypoxia or oxidative stress, MTF1 translocates into the nucleus and activates the transcription of genes bearing one or multiple copies of metal responsive elements (MREs) in their promoters, including transporters of metals and endogenous metal scavengers called metallothioneins (MTs) and other genes involved in protection from oxidative stress[61–63].

We analysed the cellular localisation of MTF1 in Q128_Mtf1 cells, and found a strong nuclear localisation (Fig. 5a), indicating that MTF1 might be directly regulating nuclear gene expression.

Thus, we performed RNAseq to identify the transcriptional program controlled by *Mtf1* conferring protection against mHTT. We first asked whether the genes, regulated by mHTT (Fig. 1f), were affected by *Mtf1* and observed that 36.8% were significantly rescued (Fig. 5b), including the HD-related genes *Creb3l3*, *Cox7b2*, and *Tspo*.

Among the biological processes altered by mHTT (Fig. 1g), MTF1 rescued glutathione, neutral amino acid, steroid and iron metabolism, as reported in literature[60,64–66]. Processes linked to histone modifications were not rescued (Fig. 5c). Of note, the number of DEGs controlled by MTF1 is much higher (1042 vs 375) than those regulated by mHTT (Supplementary Fig. 5a), indicating that *Mtf1* overexpression

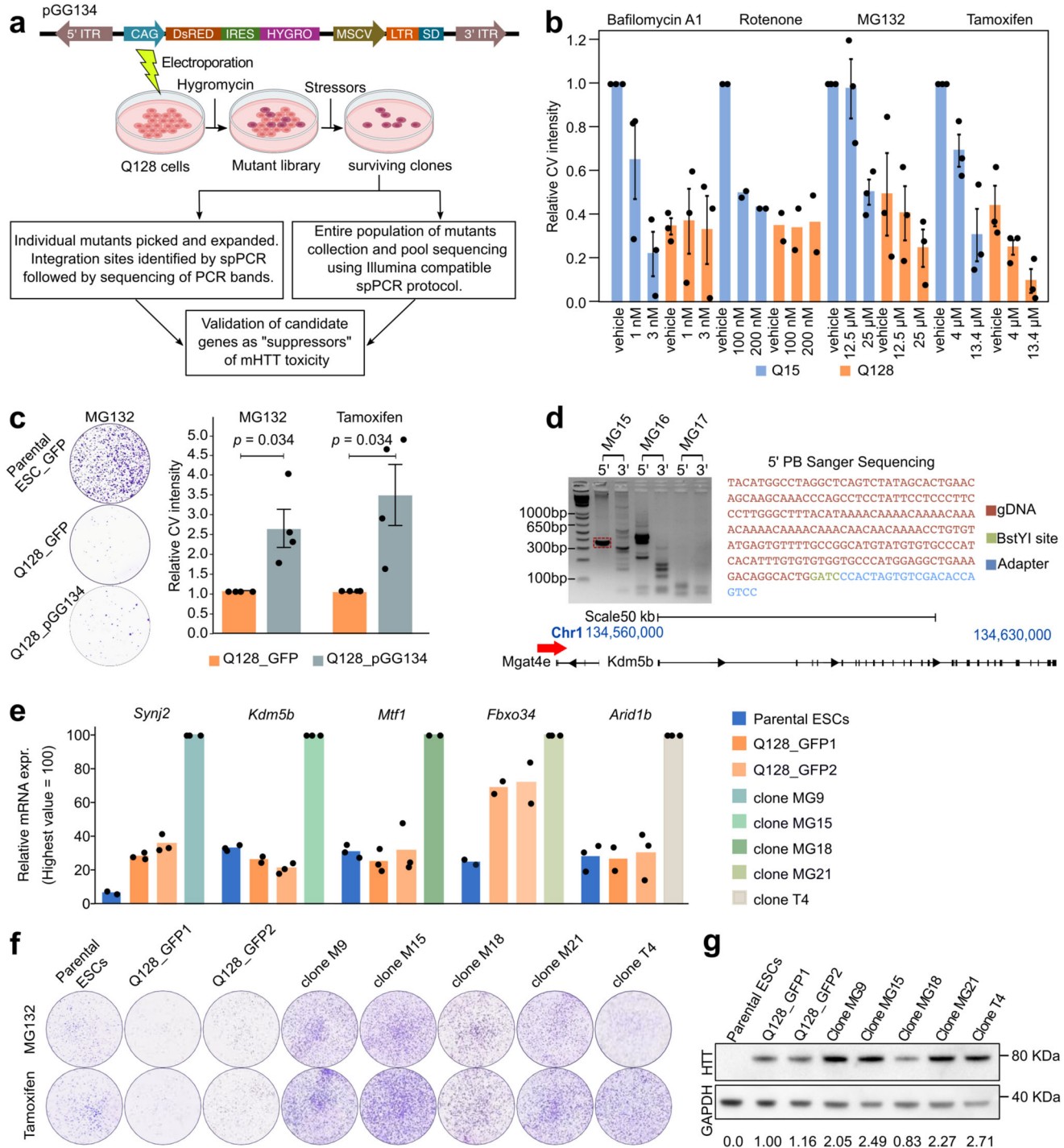

**Fig. 2 | A gain-of-function screen for suppressors of mHTT toxicity. a** Top: diagram of the PB vector pGG134[54]. Bottom: diagram of the screening strategy used to identify proteins involved in HTT-dependent toxicity. Created with BioRender.com. **b** Number of surviving cells scored by quantification of Crystal Violet (CV) staining-positive colonies upon treatment for 48 hours with the selected compounds. Bars indicate the mean ± SEM of at least 2 independent experiments, shown as dots. Data were normalised to Q15 vehicles samples. **c** Left: representative images of parental mouse ESCs or Q128 cells electroporated with PB vector encoding for GFP and Q128 cells electroporated with pGG134 vector, treated with MG132 for 5 days and stained with CV. Right: intensity of CV signal of surviving Q128_pGG134 colonies compared to Q128_GFP after mutagenesis and selection in presence of MG132 or Tamoxifen. Bars indicate the mean ± SEM of 4 independent experiments, shown as dots, normalised to Q128_GFP. P-values were calculated with

a one-tailed one sample Mann-Whitney U test. **d** Steps for the identification of integration site in the MG15 clone by Splinkerette-PCR. The band corresponding to amplification of 5′ end (red dashed square) of the pGG134 in MG15, was excised, purified and sequenced. Each clone was analysed at least two times independently. The sequence obtained (top right) was then aligned to the mouse genome, allowing identification in the MG15 clone of PB vector insertion upstream of the *Kdm5b* gene (bottom panel). **e** Expression analysis by qPCR of *Synj2*, *Kdm5b*, *Mtf1*, *Fbxo34* and *Arid1b* genes. Bars indicate the mean of 3 technical replicates shown as dots. Expression was normalised to the highest value. **f** Representative CV staining images of Q128 cells and 5 clones selected from Q128_pGG134 mutant population. **g** Western Blot confirmed HTT protein presence in all clones. Values shown below each clone are the mean of HTT intensity normalised to GAPDH intensity, used as loading control. Similar results were obtained in two additional technical replicates.

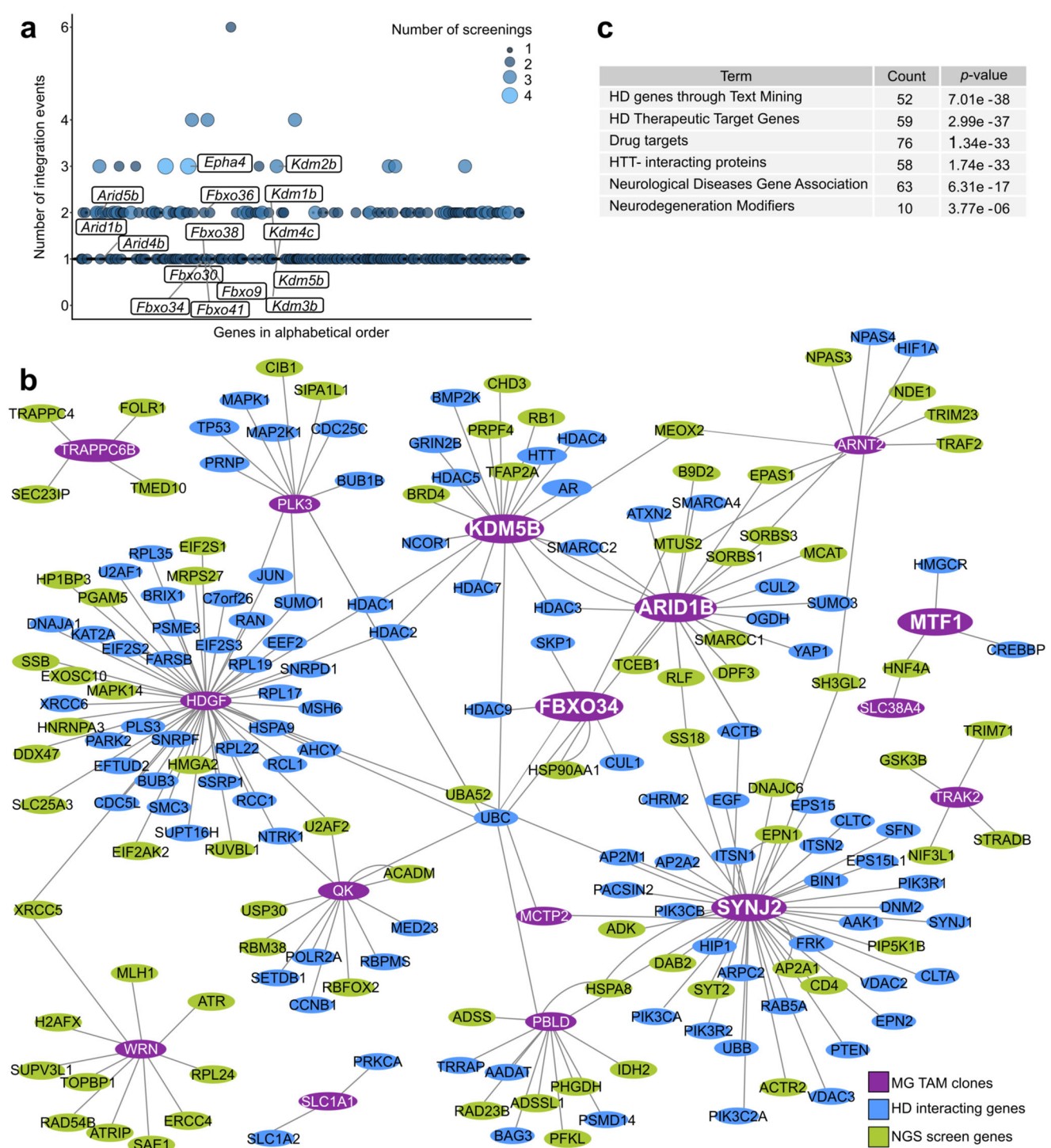

**Fig. 3 | Network analysis of candidate suppressors of mHTT toxicity. a** Bubble plot representing the number of independent integration events for each NGS hit (1356 hits, represented by bubbles and ordered along the x axis in alphabetical order). Size and colours of the bubbles are related to the number of independent screening experiments in which they were found. Labels indicate relevant candidate genes and additional genes belonging to the same families. **b** The interactome, realised with Cytoscape[141], shows physical and regulatory interactions of over-activated genes in mHTT-resistant clones with HD related genes, retrieved through HDNetDB[56], and genes identified from NGS screening. Main nodes correspond to the mutated genes of HTT-resistant clones (violet label), HD related genes are marked with a blue label and hits from the NGS screening are represented with green labels. **c** Enrichment analysis of the genes represented in the interactome (Fig. 3b) based on HD gene sets using HDNetDB. Statistical enrichment and *p*-values were calculated using the hypergeometric distribution.

drives a separate robust transcriptional program, rather than simply counteracting the changes induced by mHTT.

Therefore, we analysed the global transcriptional response elicited by *Mtf1* in Q128 cells and found that *Mt1* and *Mt2*, together with *Mtf1* itself, were the genes most significantly upregulated (Fig. 5d).

Enrichment analysis on genes regulated by MTF1 in Q128 cells revealed regulation of metal homoeostasis, regulation of steroids and glutathione (Supplementary Fig. 5b). Importantly, MTF1 did not induce any oncogenic genetic signature and did not affect any regulators of cell proliferation or apoptosis (Supplementary Fig. 5b-c), suggesting

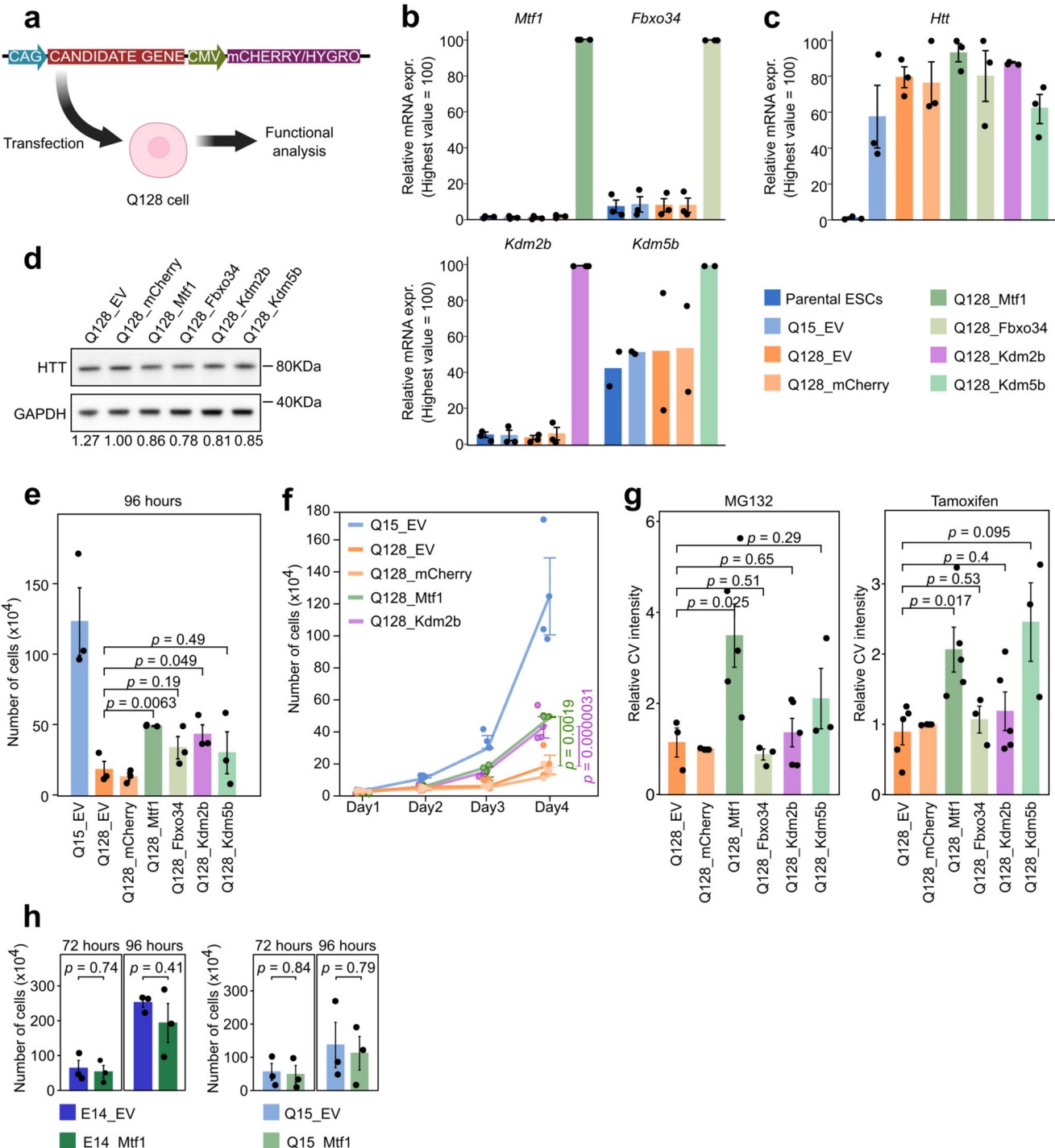

**Fig. 4 | Secondary validation of mHTT suppressors. a** Diagram of the secondary validation experiments: the cDNA of candidate genes was stably expressed in Q128 cells. An empty vector (EV) and a vector containing only mCherry cDNA served as negative controls. Created with BioRender.com. **b** Gene expression analysis by qPCR of *Mtf1*, *Kdm2b*, *Kdm5b* and *Fbxo34* confirmed increased levels of genes in corresponding cell lines in which they were overexpressed. Bars indicate the mean ± SEM of 3 independent experiments (*Mtf1*, *Fbxo34*, *Kdm2b*) and 2 independent experiments (*Kdm5b*) shown as dots. Expression was normalised to the highest value. **c** qPCR analyses for *HTT* mRNA. Bars indicate the mean ± SEM of 3 independent experiments shown as dots. Expression was normalised to the highest value. **d** Western Blot analyses of HTT protein in Q128 cells transfected with different constructs. GAPDH was used as loading control. Values shown below lane are the mean of n = 3 technical replicates of HTT intensity normalised to GAPDH intensity. **e** Proliferation assay results at 96 hours of the indicated cell lines. Bars

indicate the mean ± SEM of 3 independent experiments, shown as dots. *P*-values were calculated with unpaired two-tailed *t*-test, comparing each candidate to the Q128_EV sample. **f** Proliferation assay of the indicated cell lines. Bars indicate the mean ± SEM of 3 independent experiments, shown as dots. *P*-values were calculated with Two-way Repeated Measure ANOVA, comparing each candidate to the Q128_EV sample. **g** CV quantification showing the number of surviving colonies in Q128_Mtf1, Q128_Kdm5b and Q128_Fbxo34 cells after 48 hours of treatments with MG132 (left panel) or Tamoxifen (right panel), compared to the Q128 cell lines. Bars indicate the mean values ± SEM from at least 3 independent experiments. *P*-values were calculated with unpaired two-tailed *t*-test, comparing each candidate to the Q128_EV sample. **h** Proliferation assay at 72 and 96 hours for parental ESCs (E14) and Q15 cells, expressing either an EV or *Mtf1*. Bars indicate the mean ± SEM of 3 independent experiments, shown as dots. *P*-values were calculated with unpaired two-tailed t-test, comparing each candidate EV sample.

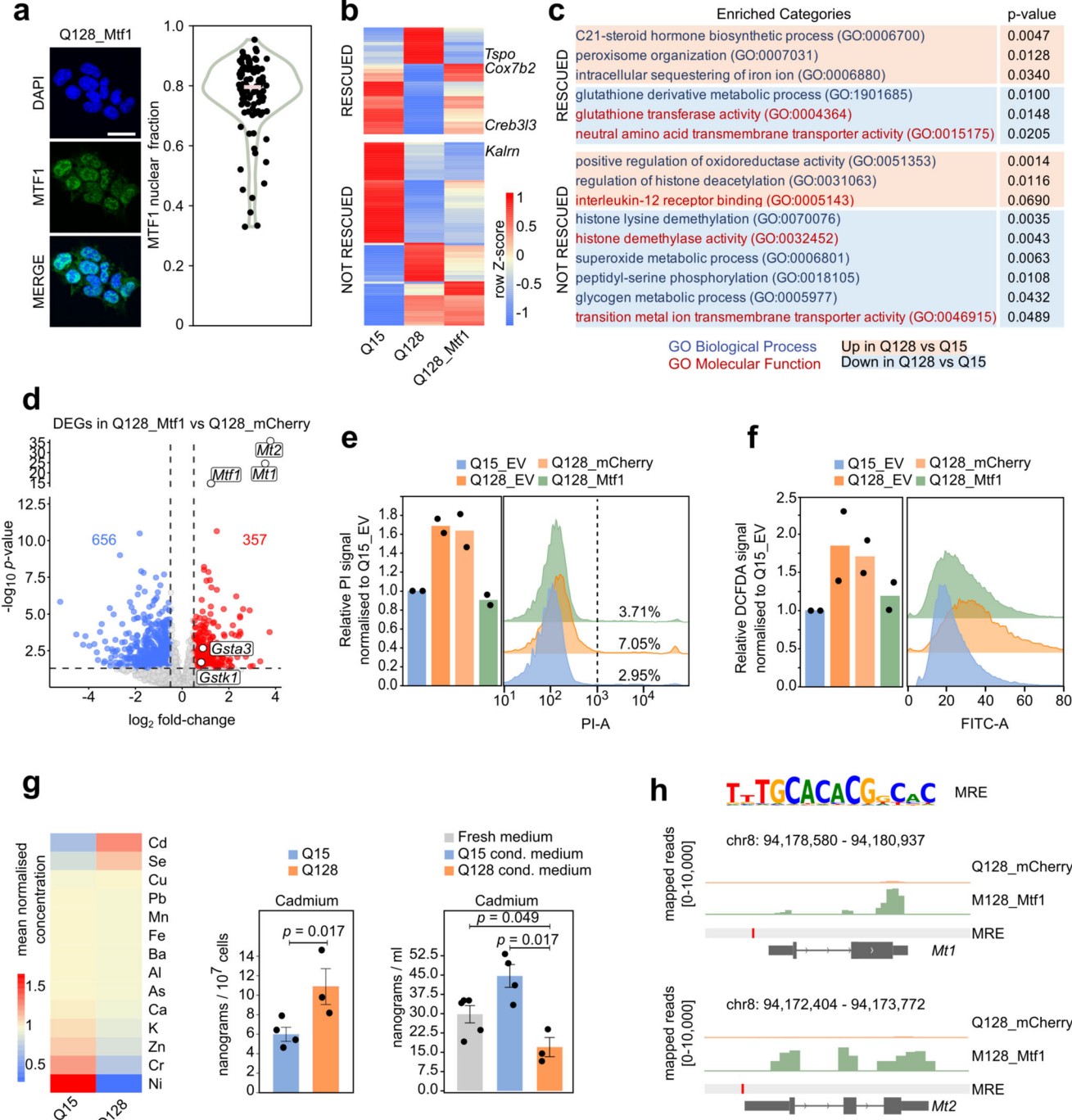

**Fig. 5 | Mtf1 regulates HD-related processes. a** Left: Immunofluorescence for MTF1 in Q128_Mtf1 cells, representative of n = 3 experiments. Nuclei were stained with DAPI. Scale bar, 30 μm. Right: Violin plot showing the fraction of MTF1 nuclear signal over the total signal from 1 experiment. **b** Heatmap showing the effect of *Mtf1* overexpression on DEGs between Q128 and Q15 cells. MTF1 rescued 138 genes (*p*-value ≥ 0.05 between Q128_Mtf1 and Q15 and log₂ FC > |0.5| between Q128_Mtf1 and Q128). Mean expression of n = 6 (Q15), n = 7 (Q128), and n = 6 biological replicates (Q128_Mtf1). Z-score calculated on row-scaled expression values. **c** Processes altered by mHTT and enriched among genes rescued, or not rescued, by MTF1. *P*-values were calculated by two-tailed Fisher Exact test using Enrichr software. **d** RNAseq on Q128_Mtf1 and Q128_mCherry cells. DOWN-regulated (log₂ FC < −0.5 and *p*-value <0.05) and UP-regulated (log₂ FC > 0.5 and *p*-value <0.05, dashed lines) genes are indicated in blue and red, respectively. *P*-values were calculated with two-tailed Wald Test with no adjustment. N = 7 (Q128_mCherry) and n = 6

biological replicates (Q128_Mtf1 cells). **e** Measurement of cell death by PI uptake and Flow Cytometry. Bars indicate the mean of 2 independent experiments, shown as dots. Representative profiles are shown on the right. **f** Measurement of ROS production by H₂DCFDA staining. Bars indicate the mean of 2 independent experiments shown as dots. Representative profiles are on the right. **g** Cellular uptake of metals in Q15 and Q128 cell lines. Left: heatmap of mean-normalised intracellular concentrations. Measurements of total intracellular (middle) and extracellular (right) amount of Cadmium in Q15_HTT and Q128_HTT cells. Fresh medium served as negative control. Mean and SEM of at least 3 biological replicates, shown as dots. *P*-values calculated with the unpaired two-tailed Mann-Whitney U test. **h** Top: consensus sequence recognised by MTF1 from Jaspar database, identified in MREs. Bottom: gene tracks for *Mt1* and *Mt2*, showing the expression levels in Q128_Mtf1 and Q128_mCherry cells; one representative biological replicate of RNA sequencing data is shown for each cell line.

that MTF1 does not act as an oncogene and it does not confer generic protection from apoptosis. To further test whether MTF1 exerts a general protective effect, independently from mHTT, we performed transcriptome analysis on Q15_Mtf1 cells. We identified genes regulated by MTF1 in Q15 cells and in Q128 cells (Supplementary Data 2). Among 1571 genes were differentially regulated in either Q15 or Q128 cells, the fraction of those regulated in both cell lines was significantly underrepresented (395 genes, p-value = 7.1e-53 Binomial test, Supplementary Fig. 5d). Consistently, only some processes (e.g. metal and steroids homoeostasis) were regulated by MTF1 also in Q15 cells (Supplementary Fig. 5b). Interestingly, genes controlling glutathione, a strong antioxidant metabolite, were regulated by MTF1 only in Q128 cells. MTF1 did not regulate genes associated with proliferation or apoptosis in Q15 cells (Supplementary Fig. 5c). Indeed, no significant effects on cell proliferation rate or protection from stress were observed upon *Mtf1* overexpression on parental mouse ESCs or Q15 cells (Fig. 4h and Supplementary Fig. 4c-d). We concluded that in mouse ESCs expressing mHTT, MTF1 partially rescues transcriptional alterations induced by mHTT, but also controls genes involved in glutathione, steroids and metal homoeostasis.

Q128 cells displayed increased ROS production and cell death (Fig. 1d-e). Thus, we asked whether *Mtf1* overexpression could prevent those processes. Notably, we detected reduced cell death (Fig. 5e) and lower ROS production (Fig. 5f) in Q128_Mtf1 compared to Q128 cells.

MTF1 activates in response to metal levels alterations[60,66,67]. We thus asked whether mHTT might alter metal homoeostasis in mouse ESCs, as suggested by the enrichment analysis on genes regulated by mHTT (Fig. 1g). We thus cultured Q15 and Q128 cells for 48 hours and performed Inductively Coupled Plasma Optical Emission Spectroscopy to measure the concentration of a panel of metals in the cells. We observed a strong increase in the concentration of Cadmium and Selenium in Q128 cells relative to Q15 cells (Fig. 5g, left). As an independent confirmation, we also measured the concentration of metals in media in which the cells were cultured, as well as in fresh medium, and found the Cadmium concentration was significantly lower in conditioned medium from Q128 cells (Fig. 5g, right). These results indicate that mHTT promotes the accumulation in mouse ESCs of Cadmium from the medium. Of note, a similar increase in intracellular levels of Cadmium has been observed also in striatal cells expressing mHTT[68,69] and it was linked to increased ROS production and cytotoxicity.

In Q128 cells we observed increased ROS production (Fig. 1e) and accumulation of Cadmium (Fig. 5g), two stimuli that could activate MTF1. However, we measured by qPCR and RNAseq the expression levels of *Mt1* and *Mt2*, as a proxy of *Mtf1* activity (Supplementary Fig. 6a and Supplementary Data 1) and found no significant differences in Q128 cells compared to Q15 cells, indicating that endogenous *Mtf1* is not activated in response to cytotoxic effects caused by mHTT. Indeed, endogenous MTF1 protein was barely detectable in Q128 cells (Supplementary Fig. 6b). In contrast, Q128_Mtf1 cells displayed a highly robust induction of *Mt1* and *Mt2* (Supplementary Fig. 6a), associated with lower ROS production and reduced cell death (Fig. 5e-f), and MTF1 showed a strong nuclear signal (Fig. 5a and Supplementary Fig. 6b). We analysed the promoters of all genes regulated by MTF1 and found that a significant fraction (11.83% p-value 1.54e-4 Fisher's exact test) bears a MRE (Fig. 5h). This evidence, together with nuclear localisation of MTF1 (Fig. 5a), indicate a direct transcriptional regulation by MTF1.

Overall, these results indicated that the presence of mHTT causes increased Cadmium accumulation, ROS production and cell death. The endogenous *Mtf1* pathway is not active, while expression of exogenous *Mtf1* is sufficient to counteract those effects, potentially through direct transcriptional regulation of MRE-containing genes.

Our screening strategy identified several suppressors of mHTT toxicity. For example, both MTF1 and KDM2B increased the number

of Q128 cells (Fig. 4e-f). We thus asked whether the combined expression of both suppressors could have synergistic effects. We generated mouse ESCs stably expressing either (Q128_Mtf1 and Q128_Kdm2b) or both candidates (Q128_Mtf1+Kdm2b). We verified that the expression of transgenes was comparable in all lines generated (Supplementary Fig. 6c). As both MTF1 and KDM2B are transcriptional regulators, we measured their effects by transcriptomic analysis in Q128 cells. MTF1 rescues a fraction (36.8%) of DEGs regulated by mHTT. KDM2B rescued a similar fraction (39.7%), while their combined expression did not increase the numbers of DEGs (37.6%). An independent analysis on the magnitude of either gene up- or down-regulation by mHTT, confirmed the lack of synergy between the two suppressors (Supplementary Fig. 6d-e and Supplementary Data 3). We conclude that MTF1 and KDM2B have similar effects on the transcriptome of Q128 cells and that they do not act synergistically in this context.

### *Mtf1* counteracts mHTT effects in zebrafish

Zebrafish is a powerful vertebrate model system widely used for human disease modelling, including HD[21,28]. Therefore, we decided to test whether *Mtf1* would display protective effects in vivo using a Zebrafish HD model. We confirmed the presence of a *Mtf1* orthologue in zebrafish, displaying high protein sequence conservation with mouse and human MTF1 (Supplementary Fig. 7a). We expressed in vivo Q16 and Q74 *HTT* fused in frame with eGFP[70] through mRNA microinjection in fertilised eggs at the one-cell stage and characterised the phenotype of injected embryos (identified by GFP expression, Supplementary Fig. 7b) at 24 hours post-fertilisation (hpf) (Fig. 6a).

Injection of Q74eGFP at titrated doses (Supplementary Fig. 7c) led to malformations as reduced body length and loss of cephalic structures. We divided the injected larvae into three main groups based on the severity of their phenotype (representative images of each group are shown in Fig. 6b, bright field panels) as Healthy embryos (H), embryos with Mild malformations (M), when embryos showed decreased head volume or cyclopia, a reduced body axis length and a curly tail, and embryos with Severe malformations (S), when embryos appeared as disorganised masses without distinguishable anterior and posterior regions. Finally, some embryos underwent rapid degeneration and widespread death and were classified as dead. Quantification of 8 independent experiments showed high percentages of mild, severe malformation and dead larvae upon Q74eGFP injections, while over 90% of Q16eGFP injected embryos were healthy (Fig. 6c), despite the same levels of the two mRNAs being detected in embryos (Fig. 6d). We concluded that only expression of mHTT in Zebrafish embryos impaired embryonic development of anterior structures.

Next, we asked whether embryonic degeneration involved cell death. Toward this aim, we performed Acridine Orange staining in 24 hpf microinjected embryos and detected regions of intense signal specifically in Q74eGFP embryos showing malformations (Fig. 6b, bottom panels). Increased cell death in Q74eGFP injected embryos was confirmed also by in situ terminal deoxynucleotidyl transferase (TdT) - mediated dUTP nick-end labelling (TUNEL) assay followed by confocal microscopy analysis and quantification. Control embryos injected with Q16eGFP mRNA showed TUNEL positivity (Fig. 6e) in body areas where apoptosis physiologically takes place at this stage of development, such as the optic vesicle, the diencephalon and the telencephalon[71]. Conversely, Q74eGFP injected embryos displayed a widespread strong TUNEL-positive signal (Fig. 6e-f) especially in the severely misshapen anterior regions (64.2% of injected embryos, with a ~ 5-fold increase in TUNEL-positive area). We concluded that microinjection of mHTT in Zebrafish embryos leads to widespread cell apoptosis.

We then asked whether *Mtf1* could suppress the detrimental effect of mHTT. Injection of *Mtf1* mRNA alone did not affect embryonic development (27/27 healthy embryos). We co-injected

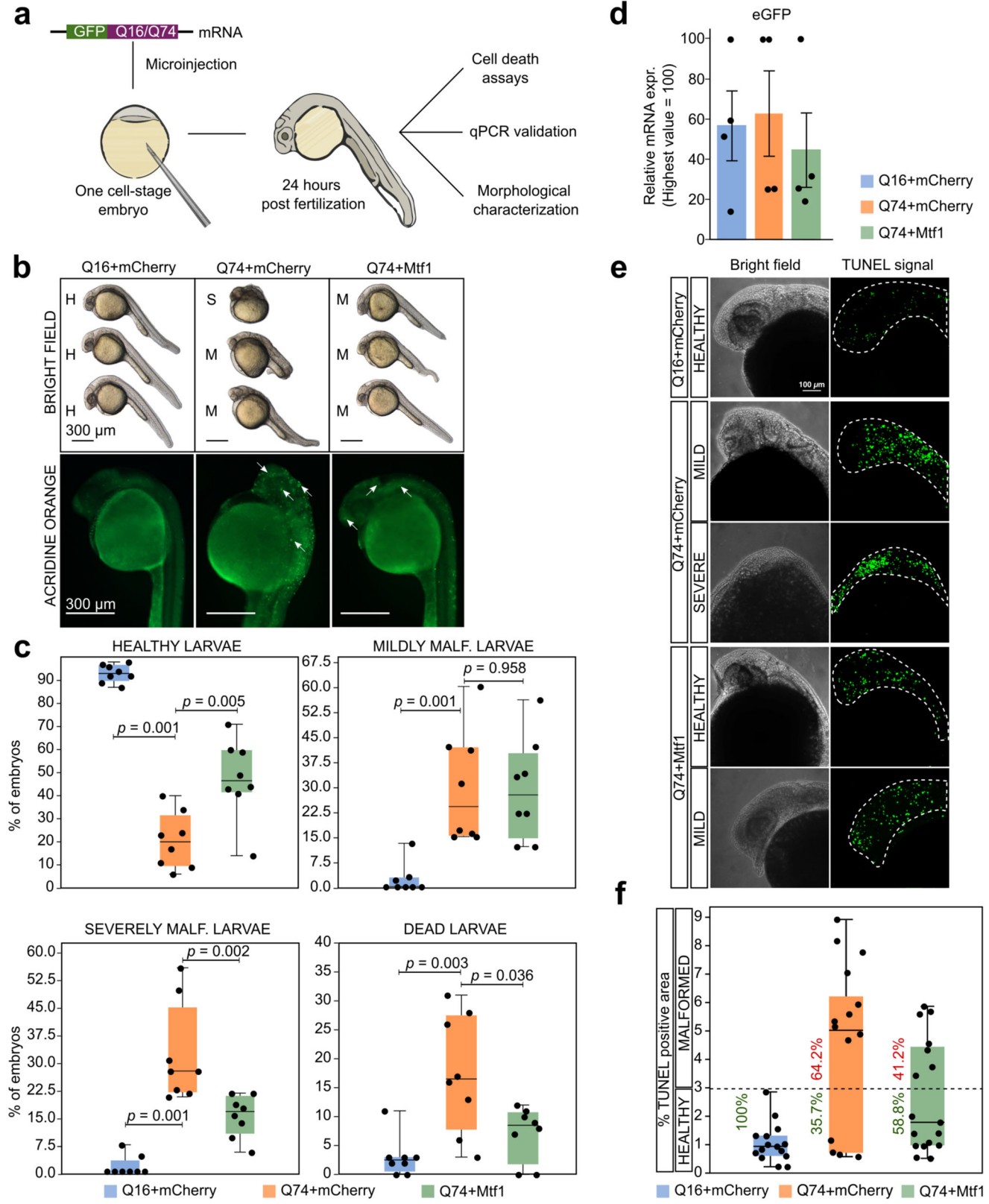

Q74eGFP and *Mtf1* mRNAs and observed a reduction in the fraction of severely malformed (from 28% to 17%) and dead embryos (from 16.5% to 8.5%, Fig. 6c), ultimately doubling the fraction of healthy embryos (from 20% to 46.5%). Crucially, *Mtf1* expression led also to a marked decrease in Acridine orange signal intensity (Fig. 6b, bottom panels) and in TUNEL-positive areas (Fig. 6e-f), indicating decreased cell death.

**AAV-vector delivery of *Mtf1* alleviates motor deficit in R6/2 mice**

Observing protective effects of *Mtf1* in a vertebrate model prompted us to test its function also in a more established HD model, such as the widely used R6/2 mice[29]. The R6/2 mice display early HD-related phenotypes characterised by locomotor hyperactivity and learning impairment (roughly 3 weeks of age)[72], followed by a progressive neurological degeneration leading to full manifestations around 8-15

**Fig. 6 | Mtf1 counteracts mHTT effects in zebrafish. a** Schematic representation of validation experiments performed in Zebrafish HD model obtained by injection of Q16eGFP or Q74eGFP mRNAs into the yolk of one cell-stage embryos. 24 hpf, embryos were collected for phenotypic and molecular analyses. **b** Representative images of 24 hours-stage embryos, injected with either Q16eGFP+mCherry, Q74eGFP+mCherry or with Q74eGFP+Mtf1 (250 pg/embryo + 250 pg/embryo), stained with the Acridine Orange. Top: Bright-field images show the morphology of representative injected embryos. Bottom: fluorescent microscopy shows increased Acridine Orange positive foci (white arrows). Images are lateral views with anterior at the top. Same results were obtained in 8 independent injection experiments. **c** Percentage of dead, malformed (severe or mild) and healthy embryos counted 24 hpf in 8 independent injection experiments, shown as dots. A total of 407, 511 and 621 embryos were analysed for Q16+mCherry, Q74+mCherry and Q74+Mtf1 samples, respectively. Box plots indicate 1st, 2nd and 3rd quartile; whiskers indicate minimum and maximum. *P*-values were calculated with the unpaired two-tailed Mann-Whitney U test. **d** eGFP gene-expression analysis by qPCR of Zebrafish embryos microinjected with eGFP and Q74+Mtf1 mRNAs. Bars indicate the mean ± SEM of 4 independent experiments shown as dots. Expression was normalised to the highest value. **e** Representative images of TUNEL assay on 30 hours-stage embryos from two independent experiments, injected with either Q16+mCherry or with Q74+mCherry or with Q74+Mtf1. Multiple focal planes were scanned for each embryo, spanning the entire depth of anterior structures, and z-projections were obtained on either bright-field and fluorescence channels. Q16+mCherry injected embryos revealed some basal TUNEL positivity, due to physiological apoptotic-dependent remodelling occurring at this stage of development. **f** Quantification of the percentage of TUNEL positive area over the total area (excluding the yolk region). Each dot represents an embryo (Q16+mCherry=15, Q74+mCherry=14, Q74+Mtf1 = 17). Box plots indicate 1st, 2nd and 3rd quartile; whiskers indicate minimum and maximum. The percentages of malformed and healthy embryos are shown in red and green, respectively. Healthy embryos are characterised by reduced TUNEL signal.

weeks with severe motor coordination deficits[73]. Such alterations are characterised by cell loss in the striatum and overall brain atrophy[29,74].

To express *Mtf1* in the mouse brain, we used AAV-PHP.eB, a capsid that has been engineered to efficiently cross the blood-brain barrier (BBB) upon intravenous injection[75,76]. This viral vector diffuses over large neural areas including basal ganglia, resulting in transduction of >90% of neurons in the striatum upon a single administration in several mouse models[77]. We first assessed whether the AAV-PHP.eB vectors we chose could efficiently cross the BBB of R6/2 mice. To this aim, 11 week old R6/2 mice were tail-vein injected with AAV-containing GFP and the viral expression was confirmed by immunoblotting. (Supplementary Fig. 8a).

Next, R6/2 mice and WT littermates underwent to a single tail-vein injection of AAV-PHP.eB packaging either GFP (AAV-GFP), used as control, or *Mtf1* (AAV-Mtf1), and motor performance was assessed weekly by Rotarod and Horizontal Ladder Task (HLT) as a functional readout of striatal neuronal loss[20,78] (Fig. 7a). R6/2 mice injected with AAV-GFP fell more rapidly than WT littermates in the Rotarod test from 7 weeks of age, as previously reported[20]. R6/2 mice injected with AAV-Mtf1 maintained performances similar to WT littermates for the entire duration of the analysis (Fig. 7b and Supplementary Fig. 8b). The HLT revealed an increased number of errors for R6/2 mice relative to WT mice at 7 weeks of age, as previously reported[79]. Injection with AAV-Mtf1 rescued such effects (Fig. 7c). An additional test of motor performance, the clasping test, further confirmed the protective effects of MTF1 (Fig. 7d). Finally, we detected reduced body weight of R6/2 mice compared to WT littermates when AAV-GFP was injected. *Mtf1* had no significant effect on the body weight of R6/2 mice (Supplementary Fig. 8c). Brain weight, an indirect measure of brain atrophy, was significantly reduced in R6/2 mice[80] transduced with AAV-GFP, relative to WT littermates (Fig. 7e), while the brain weight of R6/2 mice treated with AAV-GFP was comparable to WT. Crucially, R6/2 mice transduced with AAV-Mtf1 showed increased brain weight relative to R6/2 mice treated with AAV-GFP, indicating a protective effect of *Mtf1* (*p*-value = 0.019, unpaired two-tailed T-test).

In order to confirm expression of exogenous *Mtf1*, mice were sacrificed at 11 weeks of age and striatal and cortical tissues were collected. PCR on total DNA detected *Mtf1* viral copies in both brain regions (Supplementary Fig. 8d), thus confirming correct in vivo delivery of *Mtf1*. Altogether, these results indicated that injection of AAV-Mtf1 vector strongly and specifically ameliorated the motor defects observed in R6/2 mice.

Since MTF1 lowers ROS production in Q128 cells (Fig. 5f), we tested if AAV-Mtf1 could exert the same function in vivo. Measurement of striatal intracellular ROS levels through the dihydroethidium (DHE) fluorometric method revealed an increase in R6/2 mice relative to WT littermates. Expression of *Mtf1* restored ROS levels to those of WT animals (Fig. 7f-g). A hallmark of HD, both in patients and mouse models, is the formation of mHTT protein nuclear aggregates[81,82]. Interestingly, protein aggregates have been reported to be a consequence of oxidative stress, given that the oxidation of some amino acids induces structural changes which leads to aggregation[83,84]. We first confirmed the formation of protein aggregates in R6/2 mice, which were absent in WT littermates (Fig. 7h-i and Supplementary Fig. 8e-h). Expression of *Mtf1* significantly reduced nuclear aggregate formation in the striatum. We conclude that *Mtf1* expression improves motor function of R6/2 mice, with a concomitant reduction in molecular alterations.

## MTF1 rescues mHTT-dependent alterations in human NPCs

Our results endorse MTF1 as a factor counteracting the cytotoxic effects of mHTT. However, we used several models in which a fragment of mHTT is overexpressed, potentially leading to artefacts. Furthermore, animal models of HD might show differences in the pathogenic mechanisms, not relevant for HD therapeutic approaches. For these reasons, we decided to test the effectiveness of *Mtf1* in a human in vitro HD model, in which the endogenous *HTT* gene bears a pathogenic number of CAG repeats. NPCs can be obtained from induced pluripotent stem cells (iPSCs), and they display mHTT-specific phenotypes, such as increased ROS production and cell death[85–89]. We obtained available iPSC lines[37,90] with either 109 or 21 CAG repeats generated from patient fibroblasts or healthy controls (Fig. 8a). Both iPSCs displayed undifferentiated morphology (Fig. 8b) and robustly expressed pluripotency markers[91] (Fig. 8c and Supplementary Fig. 9a). Both lines showed comparable proliferation rates (Fig. 8d). We then generated NPCs from iPSCs ([85,92], see Methods). NPCs lost the flat epithelial morphology of iPSCs and acquired an elongated, oval shape (Fig. 8b). Molecularly, they downregulated pluripotency markers and upregulated *NES* (also known as NESTIN), *SOX1* and *PAX6*, indicating correct acquisition of neural precursor identity (Fig. 8c and Supplementary Fig. 9a-b). Both NPCs displayed comparable proliferation rates (Supplementary Fig. 9c).

We measured ROS production and observed a significant increase in NPCs bearing 109 CAG repeats (NPC_Q109), compared to control NPCs (NPC_Q21, Fig. 8e), in agreement with previous studies[85,87–89]. We reasoned that the increased levels of ROS production could make NPC_Q109 more sensitive to oxidative stress. Indeed, treatment with Rotenone induces significantly higher levels of apoptotic cells (Fig. 8f), as previously reported[85,87–89].

Finally, we asked whether expression of *Mtf1* could rescue the effects induced by mHTT in NPCs. Expression of *Mtf1* led to robust induction of a large set of MTs (Fig. 8g and Supplementary Fig. 9d). Crucially, *Mtf1* expression in NPC_Q109 reduced both ROS production and apoptosis to levels found in NPC_Q21 (Fig. 8e-f). We conclude that MTF1 counteracts mHTT-dependent cytotoxic effects also in a human HD neural model, in which mHTT is expressed at physiological levels.

 

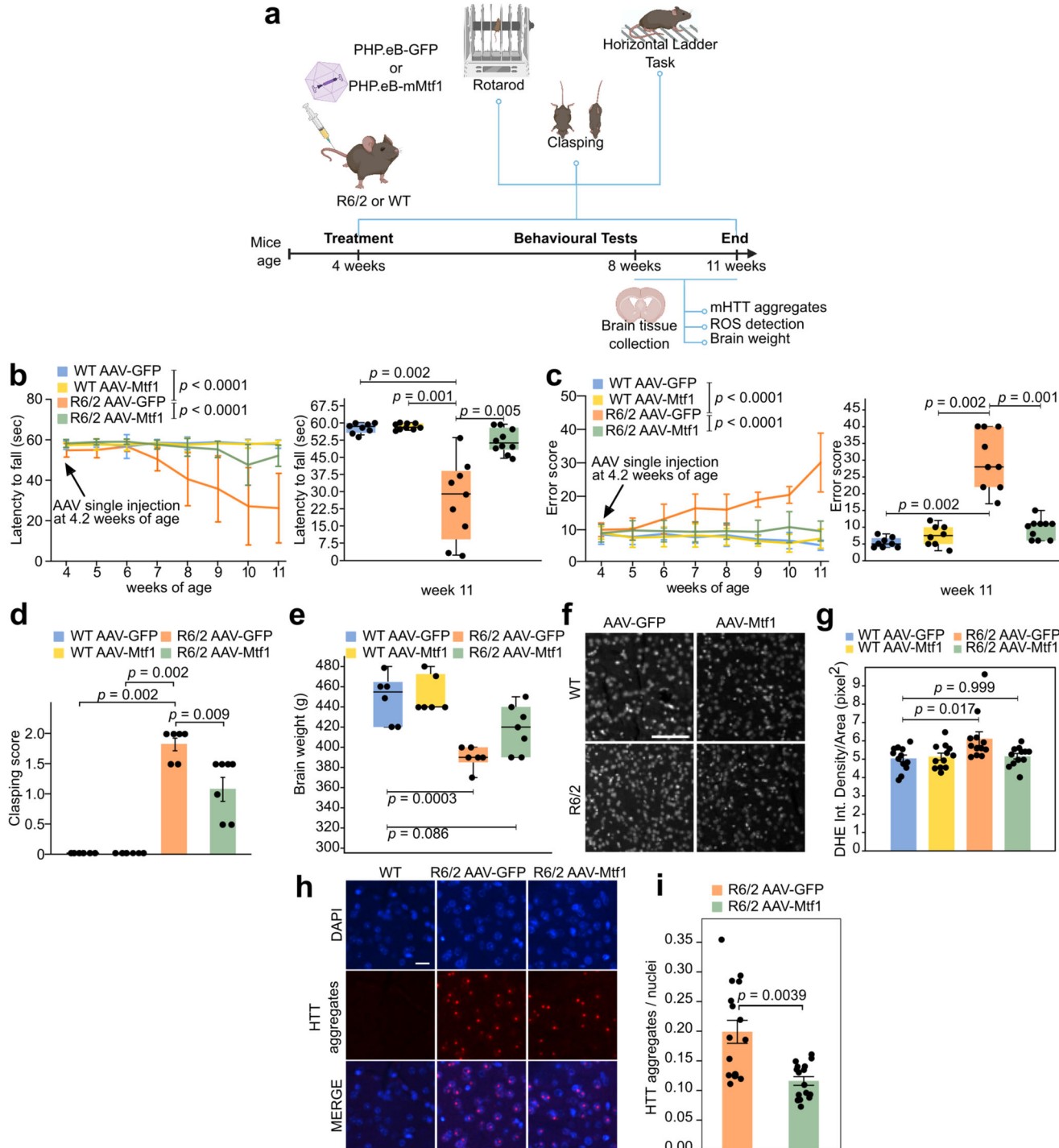

**Fig. 7 | AAV-vector delivery of Mtf1 alleviates motor deficit in R6/2 mice.**
**a** Summary of experiments performed in HD mice injected with AAVs. Adapted from "Behavioural Tests for Mice - Timeline" by BioRender.com (2023). **b** Motor performance assessed by Rotarod test. Line plots show mean ± standard deviation (SD) of each experimental group at each time point. Two-way repeated measure ANOVA. Box plots of the indicated experimental groups at 11 weeks of age. Number of mice, shown as dots: WT AAV-GFP = 8; WT AAV-Mtf1 = 9; R6/2 AAV-GFP = 9; R6/2 AAV-Mtf1 = 10. Two-tailed Mann-Whitney U test with Bonferroni correction.
**c** Analysis of motor coordination on Horizontal Ladder task. Line plots show mean ± SD. Two-way repeated measure ANOVA. Box plots of the indicated experimental groups at 11 weeks of age. Number of mice, shown as dots: WT AAV-GFP = 8; WT AAV-Mtf1 = 8; R6/2 AAV-GFP = 9; R6/2 AAV-Mtf1 = 10. Two-tailed Mann-Whitney U test with Bonferroni correction. **d** Limb-clasping response at 7-8 weeks of age. Bars indicate the mean ± SD. N = 6 mice for each group. Two-tailed

Wilcoxon test. **e** Brain weight measure. Number of mice, shown as dots: WT AAV-GFP = 6; WT AAV-Mtf1 = 6; R6/2 AAV-GFP = 6; R6/2 AAV-Mtf1 = 7. Values round off to the nearest 10 to account for minor differences in the dissection procedure. Two-way ANOVA with Bonferroni correction. **f** Detection of superoxide production by DHE. Representative images from 4 biological replicates. Scale bar, 50 μm. **g** Normalised DHE staining intensity. Bars indicate the mean ± SD. N = 4 mice for each group. N = 3 measurements for each mouse, shown as dots. Two-way ANOVA with Bonferroni correction. **h** Immunohistochemistry of the striatum of WT and R6/2 mice injected with either AAV-GFP or AAV-Mtf1 stained with EM48 antibody, which specifically detects mHTT aggregates. DAPI was used to detect nuclei. Scale bar, 10 μm. **i** Bars indicate the mean ± SD of the number of nuclear mHTT aggregates counted in 4 different levels for each animal of each group, shown as dots. N = 4 mice from each group were analysed. Two-tailed Mann-Whitney test. All box plots indicate 1st, 2nd and 3rd quartile; whiskers indicate minimum and maximum.

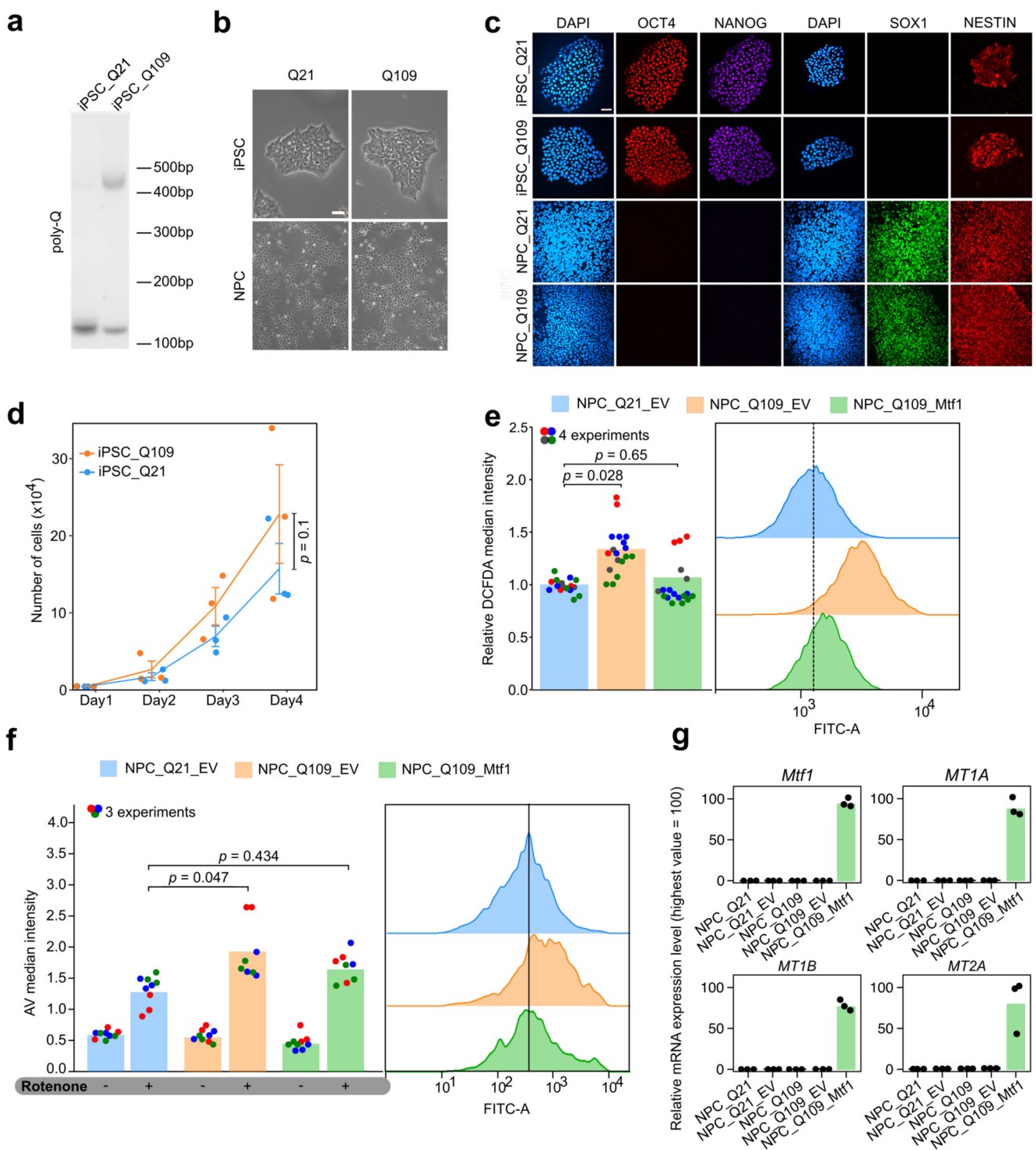

## Discussion

Several molecular pathways have been involved in the pathogenesis of HD including proteasome degradation, disruption of calcium signalling and homoeostasis, mitochondrial dysfunction, gene transcription impairment, vesicular transport and recycling alterations[26,41,51,93]. Whether these biological processes are related to each other and how they contribute to the pathogenesis of the disease is still under investigation, thus we did not focus a priori on a specific target. Instead, we performed a genome-wide screening mediated by PB transposon system[54,55] to identify genes potentially able to suppress the cytotoxic effects caused by mHTT.

Genetic screenings for HD have been carried out in model systems such as *Saccharomyces cerevisiae* and *Caenorhabditis elegans*[94,95]. These studies successfully identified therapeutic targets but suffered from limitations such as difficulty in the identification of mammalian homologues of the identified genes and the time required to generate and screening libraries of mutants. Still, these screenings have proven successful, allowing the identification of compounds that ameliorated neurodegeneration in mouse animal models of HD[96]. To overcome restrictions due to the use of model systems evolutionarily far from mammalians, we decided to carry out our PB genetic screening in mouse ESCs expressing mHTT. Our unbiased and functional approach allowed us to identify genes as potential suppressors of mHTT toxicity.

**Fig. 8 | MTF1 rescues mHTT-dependent alterations in human NPCs.**
**a** Genotyping of iPSC_Q21 and Q109 showed two amplicons for Q109 cell line (143 bp corresponds to WT allele with 19 CAG repeats, 470 bp corresponds to mutated allele with 109 CAG repeats), versus two overlapping bands in Q21 cell line. β-Actin was used as loading control. n = 1 experiment. **b** Brightfield images showing the morphology of iPSC_Q21, iPSC_Q109, NPC Q21, NPC Q109. Scale bar, 20 μm. Similar results were obtained in 7 independent experiments.
**c** Immunofluorescence for pluripotency markers OCT4 and NANOG and early neural markers SOX1 and NESTIN of iPSC_Q21, iPSC_Q109, NPC Q21, NPC Q109. Scale bar, 40 μm. Similar results were obtained in 3 independent experiments.
**d** Proliferation assay of iPSC_Q109 (orange) and iPSC_Q21 (blue) cells. Bars indicate the mean ± SEM of 3 independent experiments, shown as dots. *P*-value was calculated with Two-way Repeated Measure ANOVA. **e** Measurement of H₂DCFDA fluorescence as an evaluation of ROS production in NPC_Q21_EV versus NPC_Q109_EV, NPC_Q109_Mtf1 cells. Representative flow cytometry profiles of ROS

detection in NPC_Q21_EV, NPC_Q109_EV and NPC_Q109_Mtf1 are represented in the right panels. Bars indicate the mean of 4 independent experiments. For each experiment, technical replicates are shown as dots of different colours. FCs were calculated relative to the NPC_Q21_EV samples. *P*-values were calculated with unpaired two-tailed t-test. **f** Measurement of cell death by Annexin V uptake and Flow Cytometry. The fraction of Annexin V positive cells was calculated for each sample and FCs were calculated relative to the NPC_Q21_EV samples. Representative flow cytometry profiles of AV detection in NPC_Q21_EV, NPC_Q109_EV and NPC_Q109_Mtf1 are represented in the right panels. Bars indicate the mean of 3 independent experiments shown as dots (technical replicates of different experiments are presented with different colours). *P*-values were calculated with One-way Repeated Measure ANOVA with Tukey's correction. **g** Gene expression analysis by qPCR of *Mtf1* and metallothioneins *MT1A*, *MT1B*, *MT2A* in the indicated samples. N = 3 technical replicates, shown as dots. Expression was normalised to the highest value. Numerical values are provided in the Source data file.

---

Of note, none of them are known oncogenes. These genes belong to different biological categories, such as proteasome degradation complex, vesicular trafficking and transcriptional regulators, such as transcription factors and epigenetic modifiers, consistently with several reports showing that transcriptional deregulation is a key mechanism in HD pathogenesis as the direct interaction between mHTT and the transcriptional machinery results in abnormal mRNA expression profiles and inhibition of enzymes involved in chromatin remodelling[15,25,93,97,98].

MTF1 is itself a transcription factor and a master regulator of metal homoeostasis[60]. Upon exposure to heavy metal or other stressful conditions, as oxidative stress or infections, MTF1 translocates to the nucleus, specifically binding to MREs (metal responsive elements) and promotes transcription of metal transporters and of metallothionein genes, endogenous metal scavengers. The first MTF1 cDNA was cloned and characterised in mouse in 1993[99]. In the subsequent years, MTF1 was identified and characterised in many other vertebrate and invertebrate organisms like humans[100], fish[101,102], insects[103] and molluscs[104,105]. MTF1 is broadly characterised by an evolutionary conserved DNA binding domain composed of six zinc fingers of Cys₂His₂-type, a transcriptional activation domain[60]. The zinc finger domain is conserved in all species analysed so far and it was demonstrated that mammalian and *Drosophila melanogaster* MTF1 can cross-complement each other when tested in the respective knock-out background[106]. Similarly, expression of *Mtf1* from pufferfish *Takifugu rubripes* in *Mtf1*-null mutant mouse cells induced the transcription of a mouse *Mt1* promoter, an effect boosted by zinc and cadmium induction[67]. We expressed mouse *Mtf1* mRNA in murine, human and zebrafish models, and observed consistent protective effects and activation of target genes, such as MTs. Mouse MTF1 shows high identity of 92, 93 and 99% in the zinc finger domain with fugu, zebrafish and human (Supplementary Fig. 7a). We conclude the high evolutionary conservation of zinc finger DNA binding domain of MTF1 confers the capacity to activate target genes when MTF1 is expressed in different species, which we and others observed[67,106].

Metals such as iron, copper and zinc are essential for several cellular functions, and alterations in their levels contribute to the development of neurodegenerative diseases such as Alzheimer's disease (AD), Parkinson's disease (PD) and HD[107–110]. Iron, in particular, has been involved in various neurodegenerative diseases and in a group of disorders named neurodegeneration with brain iron accumulation[108]. Iron accumulation has been observed in different cerebral districts, such as substantia nigra in PD[111] and caudate nucleus and putamen in HD[112,113]. Iron accumulation can cause an increase in ROS production, like oxygen free radicals ($O^{2-}$) and hydrogen peroxide ($H_2O_2$), which can damage brain cells and contribute to neurodegeneration[107,114]. Moreover, metals can interact with brain proteins, promoting the formation of toxic protein aggregates which can damage neuronal cells. For example, it has been reported that β-amyloid aggregates

formation is enhanced in presence of iron in AD. Many other neurodegenerative disorders such as PD, amyotrophic lateral sclerosis, multiple system atrophy, dementia with Lewy bodies and progressive supranuclear palsy are characterised by the aggregation of insoluble protein in presence of iron[115–121]. In HD as well, metals can interact with mHTT, promoting its aggregation and increasing its toxicity[108,109,112,113]. This interaction can be mediated by ROS production, which activates cellular stress mechanisms and leads to neuronal death[107,114].

In synthesis, metals accumulation in the brain, ROS production and the formation of toxic protein aggregates have been implicated in the pathogenesis of HD. In this study, we demonstrated that forced expression of *Mtf1* rescued transcriptional alterations (Fig. 5b and c), reduced ROS production, protein aggregation caused by mHTT (Fig. 5f).

Interestingly, when *Mtf1* is expressed in healthy cells we failed to detect significant effects on proliferation or protection from cell stress/death, indicating that MTF1 does not confer a generic protection. In line with these observations, transcriptomic analyses revealed that MTF1 does not control genes involved in cell proliferation or apoptosis, but rather it controls genes associated with metal homoeostasis and protection from oxidative stress, in agreement with previous studies[61–63]. Future studies will be needed to identify the key downstream effectors of MTF1.

Metal dysregulation has been implicated in several neurodegenerative disorders, suggesting that MTF1 could work as a suppressor not only in HD. Saini and colleagues[122] reported that in *park* (also known as *parkin*) mutant flies the combined loss of *park* and MTF1 is synthetic lethal because of the increased ROS production, while the overexpression of MTF1 rescued *park* mutant phenotype acting as antioxidant. We should also stress that our genetic screening and most of our models are based on expression of exon 1 of *HTT*, which mainly encodes for the polyQ tract. Thus, the toxicity we studied is likely primarily due to the polyQ. Therefore, it will be interesting to test whether the suppressors we identified are active also against other diseases caused by polyQ expansions, such as spinal and bulbar muscular atrophy.

We delivered *Mtf1* by means of AAV-vectors, given that they have been successfully employed in clinical settings[123–126], including neurodegenerative diseases[75,77] displaying long-term and sustained effect of AAV gene therapy upon a single intravenous injection[127–129]. This approach represents a preventive strategy to delay the progression of neuronal degeneration typical of HD. In fact, we administered our treatment in the experimental cohort of R6/2 mice during the pre-symptomatic phase in order to mimic these conditions. Besides, *Mtf1* delivery by AAV-vectors well accommodates the possibility of an integrated intervention in combination with antisense oligonucleotides lowering mHTT production[130], or in combination with other gene-based strategies, such as in vivo reprogramming of glial cells into neurons[131–134] or CRISPR/Cas9-mediated inactivation of the *HTT* gene[88,135].

We are aware of the difficulties in translating our findings from animal models to the clinic. For example, the AAV-vectors that we used do not allow efficient brain transduction in primates. Thus, administration of AAV-vectors successfully used in patients[136] directly to the striatum by stereotaxis could represent a viable strategy for clinical applications. We will also investigate the possibility to activate endogenous *Mtf1* expression by small molecules.

It is indeed conceivable that future therapeutic interventions, working at different levels of HD pathophysiology, will have more chances to be successful.

## Methods

### Cell culture

Mouse ESC lines (Rex1GFP-d2 and E14IVc[137]) were cultured in feeder free conditions [plastic coated with 0.2% gelatine (Sigma, cat. G1890)] and replated every 3–4 days at a split ratio of 1:10 following dissociation with Accutase (GE Healthcare, cat. L11-007) or 0.25% Trypsin (Life Technologies). Cells were cultured in serum-free N2B27-based medium [DMEM/F12 and Neurobasal in 1:1 ratio, 0.1 mM β-mercaptoethanol, 2 mM L-glutamine, 1:200 N2 and 1:100 B27 (all reagents from Life Technologies)] or serum-containing KSR medium [GMEM (Sigma, cat. G5154) supplemented with 10% KSR (Life Technologies), 2% FBS (Sigma, cat. F7524), 100 mM β-mercaptoethanol (Sigma, cat. M7522), 1× MEM non-essential amino acids (Invitrogen, cat. 1140-036), 2 mM L-glutamine, 1 mM sodium pyruvate (both from Invitrogen)], supplemented with two small-molecule inhibitors (2i) PD (PD0325901, 1 μM), CH (CHIR99021, 3 μM) from Axon (cat. 1386 and 1408) and LIF (100 units/mL purchased from Qkine - Cambridge UK).

Human iPSCs CS09iHD-109n5 (here referred as iPSC_Q109) and CS14iCTR-21n3 (here referred as iPSC_Q21) purchased from Cedars-Sinai Biomanufacturing Center (Los Angeles, California), were maintained on pre-coated 0.5% Matrigel (CORNING, cat. 356231) plates in E8 medium made in house (according to Chen et al., 2011[138]) or in mTeSR (StemCell Technologies, cat. 05850) at 37 °C, 5% O₂, 5% CO₂. Human iPSCs were dissociated in clumps with 0.5 mM EDTA (Gibco, cat. AM99260G) and replated at 1:6 dilution every 3-4 days with 10 μM ROCK inhibitor (Y27632-dihydrochloride Axon Medchem, cat. 1683) for 24 hours. Medium was changed every day. All cell lines were mycoplasma-negative.

### Genotyping of iPSC_Q21 and iPSC_Q109

Genomic DNA from iPSC_Q21 and iPSC_Q109 was isolated using the DNeasy Blood and Tissue kit (Qiagen, cat. 69504) following manufacturer's instructions and quantified by Nanodrop ND-1000. PCR was performed using primers listed in Supplementary Table 1 and described in Mangiarini et al.[29], designed to amplify polyQ tract in HTT exon 1. For a 25 μL reaction we used 200 ng of genomic DNA, 0.25 μL of Taq Phusion High fidelity (ThermoFisher, cat. F-530L), 2.5 μL of Buffer GC 5x + MgCl² (7.5 mM), 2 μL of dNTPs (10 mM) and 1.25 μL of Dimethyl sulfoxide (DMSO, 100%). After 3 minutes at 94 °C, we performed 35 cycles of: 94 °C for 1 minute, 63 °C for 45 seconds, 72 °C for 1 minute, followed by a final elongation at 70 °C for 7 minutes. Gel electrophoresis was performed on agarose gel at 2.5%, loading 20 μL of PCR products with Purple Loading Dye 6x (NEB, cat. B7024S). β-Actin was used as loading control. Results were digitally acquired by VWR Imager CHEMI Premium.

### Generation of HTT-expressing ESC lines

Q15 and Q128 cells were generated by DNA transfection of vectors containing N-terminal of human huntingtin gene, with 128 or 15 CAG repeats respectively (courtesy of Professor Elena Cattaneo). Overnight linearization of plasmid DNA was performed with the restriction enzyme PvuI. For DNA transfection, we used Lipofectamine 2000 (Life Technologies, cat. 11668-019) and performed reverse transfection. For one well of a 6-well plate, we used 6 μl of transfection reagent, 2 μg of

plasmid DNA and 300,000 cells in 2 mL of medium. The medium was changed after overnight incubation. Antibiotic selection (Puromycin 1 μg/mL) started 24 hours after transfection.

### Generation of mouse ESCs stably expressing genes of interest

Stable transgenic mouse ESCs expressing candidates were generated by transfecting HTT-expressing cells with PB transposon plasmids (1 μg of CAG-Mtf1, CAG-Kdm2b, CAG-Kdm5b and CAG-Fbxo34), purchased from VectorBuilder (VectorBuilder Inc, Chicago, IL, USA), with PB transposase expression vector pBase (1 μg). We used Lipofectamine 2000 and performed reverse transfection as described for HD lines generation. Antibiotic selection (Hygromycin B, 150 μg/mL; Invitrogen, cat. 10687010) started 24 hours after transfection.

### NPCs differentiation

iPSC_Q21 and iPSC_Q109 were differentiated into NPCs according to Li et al., 2011[92] protocol. iPSCs at 80% confluency were dissociated in single cells with Accutase (Gibco, cat. A1110-501) and plated 45,000 cells/cm² in E8 medium with 10 μM ROCK inhibitor. After 1 day, E8 medium was substituted with N2B27 induction medium composed of Advanced DMEM/F12 (Gibco, cat. 12634-010): Neurobasal (Gibco, cat. 21103-049) (1:1 ratio), BSA 50 mg/mL (Gibco, cat. 15260-037), Glutamax 1% (Gibco, cat. 35050-038), Penicillin/Streptomycin 1% (Gibco, cat. 15140122), N2 Supplement 1:200 (Gibco, cat. 17502-048), B27 Supplement 1:100 (Gibco, cat. 17504-044), supplemented with small molecules human LIF 10 ng/mL (Qkine, cat. Qk036), SB431542 2 μM (Axon Medchem, cat. 1661), CHIR99021 3 μM (Axon Medchem, cat. 1386), Compound E 0.1 μM (Sigma-Aldrich, cat. 209986-17-4). N2B27 induction medium was changed every day, for 7 days. On day 7, NPCs were splitted at 1:6 dilution in N2B27 maintenance medium which is composed of Advanced DMEM/F12 (Gibco, cat. 12634-010): Neurobasal (Gibco, cat. 21103-049) (1:1 ratio), BSA 50 mg/mL (Gibco, cat. 15260-037), Glutamax 1% (Gibco, cat. 35050-038), Penicillin/Streptomycin 1% (Gibco, cat. 15140122), N2 Supplement 1:200 (Gibco, cat. 17502-048), B27 Supplement 1:100 (Gibco, cat. 17504-044), supplemented with small molecules human LIF 10 ng/ml (Qkine, cat. Qk036), SB431542 2 μM (Axon Medchem, cat. 1661), CHIR99021 3 μM (Axon Medchem, cat. 1386), supplemented with EGF 20 ng/mL (R&D, cat. 236-EG) and FGF2 20 ng/mL (Qkine, cat. Qk002, recombinant zebrafish FGF2). NPCs were maintained for 6 passages. h-iPSCs and NPCs morphology data were digitally collected with microscope Zeiss Axio Vert A1 FL-LED.

### Generation of NPCs transiently expressing genes of interest

For DNA transfection, 250,000 NPCs were dissociated as single cells with Accutase (Gibco, cat. A1110-501) and were transfected with PB constructs (1 μg) using FuGENE HD Transfection (Promega, cat. E2311), following the protocol for reverse transfection. For one well of a 12-well plate, we used 3.9 μL of transfection reagent, 1 μg of plasmid DNA and 250,000 cells in 1 mL of N2B27 maintenance medium medium with 10 μM Y27632 [ROCKi, Rho-associated kinase (ROCK) inhibitor, Axon Medchem cat. 1683]. The medium was changed after overnight incubation. After 48 hours post transfection, cells were treated with Rotenone 30 μM for 24 hours and then analysed as indicated in Fig. 8f.

### Proliferation assay

Mouse ESCs were conditioned for one passage in KSR + 2iL medium in presence of Puromycin 6 μg/mL. Proliferation of ESCs was assessed by plating 15,000 cells in a 24-well plate (7,500 cells/cm²) in KSR + 2iL medium in presence of Puromycin 6 μg/mL. Cells were dissociated with 0.25% Trypsin (Life Technologies) and counted every 24 hours for 4 days.

Proliferation of iPSCs was measured by plating 40,000 single cells on pre-coated 0.5% Matrigel (CORNING, cat. 356231) 12-well plates (11,428 cells/cm²) in E8 medium with 10 μM ROCK inhibitor

(Y27632-dihydrochloride, Axon Medchem, cat. 1683) for 24 hours. Medium was changed every day. Cells were dissociated with Accutase (GE Healthcare, cat. L11-007) and counted every 24 hours for 4 days.

Proliferation of NPCs was measured by plating 100,000 cells on pre-coated 0.5% Matrigel (CORNING, cat. 356231) 12-well plates (28,571 cells/cm²) in N2B27 maintenance medium with 10 µM ROCK inhibitor (Y27632-dihydrochloride Axon Medchem, cat. 1683) for 24 hours. Cells were dissociated with Accutase (GE Healthcare, cat. L11-007) and counted every 24 hours for 4 days.

### Stressors treatment and Crystal violet (CV) staining

For experiments in Figs. 2b, 5,000 mouse ESCs were plated in a 24-well plate (2,500 cells/cm²) in KSR + 2iL medium in the presence of the inhibitors (and Puromycin 6 µg/ml) for 48 hours and scored by quantification of the number of surviving cells by CV staining [CV solution: 0.05% w/v Crystal Violet (Sigma), 1% of formaldehyde solution 37% (Sigma), 1% methanol, 10% PBS] and quantification of mean intensity was performed with Fiji software (v2.0.0).

For PB-mutagenesis followed by stressor treatments, cells were plated at density 2,500 cells/cm² in Puromycin 6 µg/mL and selected for 5 days in the presence of MG132 (Sigma-Aldrich, cat. C2211) or Tamoxifen (Sigma-Aldrich, cat. T2859).

For experiments in Fig. 4g, 5,000 cells were plated in a 24-well plate in KSR + 2iL medium with Puromycin 3 µg/mL. Stressors (MG132 12.5 nM or Tamoxifen 13.4 µM) were added after 12 hours. Scoring of surviving cells was performed as described above.

For experiments in Supplementary Fig. 4d, 2,500 cells were plated in a 48-well plate in KSR + 2iL medium. Stressors [Rotenone (Sigma-Aldrich, cat. R8875), Cumene (Sigma-Aldrich, cat. 247502), 5-Azacytidine (Sigma-Aldrich, cat. A1287), MG132 (Sigma-Aldrich, cat. C2211), Bafilomycin (Sigma-Aldrich, cat. B1793), Staurosporine (Sigma-Aldrich, cat. S6942), Tamoxifen (Sigma-Aldrich, cat. T2859)] were added after 12 hours at the indicated concentrations. After 48 hours, surviving cells were stained with CV solution, scoring of surviving cells was performed as described above.

### Electroporation of the PB system in ESCs

PB-mediated mutagenesis by electroporation was performed for genome-wide screening. PB vectors integrate stably in the genome after random insertion in TTAA sites. The PB pGG134 vector used (shown in Fig. 2a) was optimised for gain-of-function screens[54]: it consists of the MSCV enhancer/promoter followed by a splice donor site from exon 1 of *Foxf2* gene, which allows the over-activation of nearby genes. The PB 5′-ITR has also weak directional promoter activity, i.e. this construct can activate genes in either orientation. The vector contains also a second cassette, including a constitutive promoter followed by DsRed and Hygromycin resistance genes, which was used to identify cells with stable vector integration.

We optimised the conditions in order to achieve a low number of integration events, by adjusting the ratio of PB vector vs transposase pBase. For the screening procedure, mutagenesis was performed using the optimised amount of 0.5 µg pGG134 and 20 µg pBase.

For a single electroporation, 10⁷ cells and 20.5 µg DNA were mixed and placed into an electroporation cuvette (Biorad Gene Pulser Cuvette, cat. 165-2088). Cells were electroporated by placing the cuvette in the electroporation holder of the Biorad GenePulser (cat. 165-2076). Settings used: 250 V, 500 µF, time constant should be between 5.6 and 7.5. Electroporated cells were gently recovered from the cuvettes and plated. Antibiotic selection started 24 hours after electroporation.

### Genomic DNA extraction and Splinkerette-PCR

Cells were harvested and incubated overnight at 56 °C with lysis buffer [10 mM Tris-HCl, pH 7.5; 10 mM EDTA; 10 mM NaCl; 0.5% w/v Sarcosyl, supplemented with proteinase K (Sigma, cat. P2308) to a final concentration of 1 mg/mL]. In order to obtain DNA precipitates, the next day 2 mL of a mixture of NaCl and ethanol (30 µL of 5 M NaCl mixed with 20 mL of cold absolute ethanol) was added. Cellular extracts were centrifuged for 45 minutes at 4 °C to remove soluble fraction. Precipitated gDNA was rinsed three times by dripping 2 mL of 70% ethanol and finally resuspended in 70 °C milliQ water.

Splinkerette-PCR procedure for PB-integration mapping was adapted from Potter and Luo[139] and consisted of the following steps: a) 2 µg of genomic DNA were digested with 10 U BstYI (10,000 U/mL, NEB) in a volume of 30 µL. Reaction was incubated at 60 °C overnight, the following day the enzyme was inactivated at 80 °C for 20 minutes. Adapters for Splinkerette-PCR were generated by annealing of 150 pmol of AdapterA and B primers (Supplementary Table 1) in a final volume of 100 µL (10x NEB Buffer 2). Oligos were denatured at 65 °C for 5 min, then cooled; b) Ligation was performed in a total volume of 6 µl including a 2x Ligation mix (Takara), 2.5 µL of digested gDNA and 0.5 µL of annealed adapters for Splinkerette-PCR. Ligation reaction was incubated at 16 °C overnight, the next day 65 °C for 10 minutes for enzyme inactivation. A purification step was included before step C, using QIaquick PCR Purification Kit, following manufacturer's instructions. For PCR amplifications we used Phusion HF DNA Pol (NEB) in 5x Phusion GC Buffer recommended in case GC-rich templates or those with secondary structures. PCR mix included 5x GC Buffer, 10 mM dNTPs, DMSO and Phusion Pol; c) First round PCR was amplified with 15 µL of ligated DNA (or 50% of ligation product for each reaction for PB5′ and PB3′ transposon/host junctions), 0.5 µM for each primer (Adaptor-PCR1 and PB5′ or PB3′-ITR PCR1), 6.5 µL PCR mix, final volume of 25 µL. Splinkerette-PCR1 program: 95 °C for 2 minutes; two cycles of 95 °C for 20 seconds, 65 °C for 30 seconds, 68 °C for 2 minutes; then 30 cycles of 95 °C for 30 seconds, 60 °C for 30 seconds, 68 °C for 2 minutes; then 68 °C for 10 minutes; d) For second round PCR, we used 5 µL of 1:500 dilution of PCR1 product, 0.5 µM for each primer (Adapter-PCR2 and PB5′ or PB3′-ITR PCR2), 6.5 µL PCR mix, final volume of 25 µL. Splinkerette-PCR2 program: 95 °C for 2 minutes; two cycles of 95 °C for 20 seconds, 65 °C for 30 seconds, 68 °C for 2 minutes; then 5 cycles of 95 °C for 30 seconds, 60 °C for 30 seconds, 68 °C for 2 minutes; then 25 cycles of 95 °C for 30 seconds, 58 °C for 30 seconds, 68 °C for 2 minutes; then 68 °C for 10 minutes; e) PCR2 products were treated with Antarctic Phosphatase and Exonuclease I (both from NEB) and sequenced using PB5′- or PB3′-ITR PCR2 primers. Primers and adaptor sequences are listed in Supplementary Table 1.

### Next Generation Sequencing analysis of genomic integration sites

Genomic DNA from entire populations of mutants was extracted using a Gentra Puregene Cell Kit. Library preparation and sequencing was performed as previously described[140]. A bespoke bioinformatics pipeline allowed to map each single read to a genomic locus and to associate each site of integration to a gene within 20 kb of distance. Data was then organised into the network of HD interacting genes by means of Cytoscape software (v3.8.2)[141].

### Propidium iodide (PI) staining

PI staining was performed on live single mouse ESCs according to the manufacturer's instructions (Ebioscience, cat. 88-8007-72). After washing in PBS, 10⁵ live cells were resuspended in 200 µL of 1x Binding Buffer and 5 µL of PI Staining Solution (cat. 00-6990) were added. Flow cytometry analysis was performed using a BD FACSCanto™ II cytometer within 1 hour, storing samples at 2-8 °C in the dark. Data were analysed with BD FACSDiva™ (v. 9.0) and FlowJo (10.8.1) software. Representative gating strategy is available in Supplementary Fig. 10a.

### ROS measurement assay

ROS production was detected by staining single live mouse ESCs and human NPCs cells with 2′,7′-dichlorodihydrofluorescein diacetate (H₂DCFDA; Life Technologies, cat. D399), performing the following

steps: a) ROS indicator was freshly reconstituted in order to make a concentrated stock solution (10 mM); b) 3-5×10⁵ cells were harvested, c) washed once with 500 μL of PBS and d) resuspend in 300 μL PBS containing the probe to provide a final working concentration of 0.5 μM dye; e) cells were incubated at 37 °C for 10 minutes in the dark; f) after removal of the staining solution, samples were g) washed twice in PBS. Samples were collected by flow cytometry using a BD FACSCanto™ II cytometer or BIO-RAD S3e Cell Sorter and analysis was performed with BD FACSDiva™ (v. 9.0), ProSort™ (v. 1.6) and FlowJo (10.8.1) software. Representative gating strategy is available in Supplementary Fig. 10b-c.

### Annexin V staining
Live NPCs, transiently transfected with the gene of interest and treated with Rotenone 30 μM for 24 hours, were stained with Annexin V according to the manufacturer's instructions (Ebioscience, cat. 88-8007-72). Cells were washed once in PBS, then once in 1x Binding Buffer (cat. 00-0055). 5×10⁵ cells were resuspended in 200 μL of 1x Binding Buffer and incubated with 5 μL of fluorochrome-conjugated Annexin V (cat. 17-8007) for 10 minutes at room temperature. Cells were then washed in 500 μL of 1x Binding Buffer. Finally, cells were resuspended in 200 μL of 1x Binding Buffer. Flow cytometry analysis was performed using the BIO-RAD S3e Cell Sorter within 1 hour, storing samples at 2-8 °C in the dark. Data were analysed with ProSort™ (v. 1.6) and FlowJo (10.8.1) software. Representative gating strategy is available in Supplementary Fig. 10c.

### Western Blotting
Cells were washed in cold PBS and harvested in lysis buffer (50 mM Hepes pH 7.8, 200 mM NaCl, 5 mM EDTA, 1% NP40, 5% glycerol), freshly supplemented with 1 mM DTT, protease inhibitor (Roche, cat. 39802300) and phosphatase inhibitor (Sigma-Aldrich, cat. P5726). Samples were exposed to ultrasound in a sonicator (Diagenode Bioruptor) and centrifuged at 15,871 rcf for 10 minutes to prepare supernatant. Protein concentration was determined by Bradford quantification. For experiments in Figs. 1b, 2g, 4d and Supplementary Fig. 1d, total protein (10 μg) was fractionated on 4-12% Nupage MOPS acrylamide gel (Life Technologies, cat. BG04125BOX/BG00105BOX) and electrophoretically transferred on a PVDF membrane (Millipore, cat. IPFL00010) in a Transfer solution (50 mM Tris-HCl, 40 mM glycine, 20% methanol, 0.04% SDS). Membranes were then saturated with 5% Non-Fat Dry Milk powder (BioRad; 170-6405-MSDS) in TBST (8 g NaCl, 2.4 g Tris-HCl, 0.1% Tween20/litres, pH 7.5) for 1 hour at room temperature and incubated overnight at 4 °C with anti-HTT (clone 1HU-4C8) or anti-GAPDH (clone 6C5) primary antibody (Supplementary Table 2). Membranes were then incubated with secondary antibodies conjugated with a peroxidase, diluted in 1% milk in TBST. Pico SuperSignal West chemiluminescent reagent (Thermo Scientific, cat. 34078) was used to incubate membranes and images were digitally acquired by ImageQuant LAS 4000 (GE Healthcare). Uncropped gels and numerical values are provided in the Source data file.

Mice were sacrificed by cervical dislocation and tissues were homogenised in lysis buffer containing 20 mM Tris-HCl, pH 7.4, 1% Nonidet P-40, 1 mM EDTA, 20 mM NaF, 2 mM Na₃VO₄ and 1:1000 protease inhibitor mixture (Sigma-Aldrich) and sonicated. For experiment in Supplementary Fig. 1b, 50 μg of total protein lysate of WT mouse, R6/2 mouse, Q15 cells and Q128 cells were loaded in a handcast 5-11% acrylamide stacking-gel and incubated with the following antibodies: anti-HTT (clone EM48) and anti-GAPDH (Supplementary Table 2). For experiments in Supplementary Fig. 8a, d, e and g, 40 μg of total protein lysate were immunoblotted with the following antibodies: anti-GFP, anti-ACTIN (clone 8H10D10), anti-HTT (clone EM48), anti-TUBULIN (clone B-5-1-2) (Supplementary Table 2). Membranes were processed as described above. Images were digitally acquired by VWR Imager CHEMI Premium or ChemiDoc XRS+ (model n°: Universal Hood

II) with Image Lab Software, BioRad (v. 6.1). Quantification was performed using Fiji 2.9.0 with background subtraction and normalising on the loading control (GAPDH, ACTIN or TUBULIN). Uncropped gels are provided in the Source data files.

### Immunofluorescence
Immunofluorescence for MTF1 detection in mouse ESCs was performed on cells plated on fibronectin (Merck, cat. FC010)-coated glass coverslips. Cells were fixed in 4% formaldehyde (Sigma-Aldrich, cat. F8775) in PBS for 10 minutes at room temperature, washed in PBS, permeabilised for 10 minutes in PBS + 0.1% Triton X-100 (PBST) at room temperature and blocked in PBST + 3% of horse serum (HS; Gibco, cat. 16050-122) for 45 minutes at room temperature. Cells were incubated overnight at 4 °C with primary antibodies (for primary and secondary antibodies details see Supplementary Table 2) in PBST + 3% of HS. After washing with PBST, cells were incubated with secondary antibodies (Alexa, Life Technologies) for 45 minutes at room temperature. Nuclei were stained with DAPI (4′,6-diamidino-2-phenylindole; Sigma-Aldrich, cat. F6057). Images were acquired with Leica SP5 confocal microscopes. Image analysis was performed using Fiji 2.9.0. Quantification of nuclear and cytoplasmic intensity was performed with CellProfiler software (v4.1.3). Briefly, nuclei were detected by DAPI staining. Cytoplasm was arbitrarily defined by radially expanding nuclei by 10 pixels. Ratio of the integrated nuclear intensity of MTF1 signal over the cellular integrated intensity of MTF1 signal (nucleus + cytoplasm) was calculated and plotted.

Immunofluorescence analysis on human iPSCs and NPCs was performed on 1% Matrigel-coated glass coverslips in wells. Cells were fixed in 4% formaldehyde (Sigma-Aldrich, cat. F8775) in PBS for 10 minutes at room temperature, washed in PBS, permeabilised for 1 hour in PBST at room temperature and blocked in PBST + 5% of HS (Gibco, cat. 16050-122) for 5 hours at room temperature. Cells were incubated overnight at 4 °C with primary antibodies (for primary and secondary antibodies details see Supplementary Table 2) in PBST + 3% of HS. After washing with PBS, cells were incubated with secondary antibodies (Alexa, Life Technologies) for 45 minutes at room temperature. Nuclei were stained with DAPI (4′,6-diamidino-2-phenylindole; Sigma-Aldrich, cat. F6057). Images were acquired with Zeiss LSM900 Airyscan 2 confocal microscopes. Image analysis was performed using Fiji 2.9.0.

### RNA isolation, reverse transcription and quantitative PCR
For cellular lysate, RNA was isolated using Total RNA Purification Kit (Norgen Biotek, cat. 37500) and complementary DNA (cDNA) was made from 500 ng using M-MLV reverse transcriptase (Invitrogen, cat. 28025-013), RNaseOUT (40 units/μL), random primers (200 nM), dNTPs (10 mM), First-Strand Buffer 5x, DTT (0.1 M). For zebrafish larvae, total RNA was isolated taking advantage of the phenol-chloroform extraction. Total RNA was isolated from pools of 10 animals by using TRIzol Reagent (Life Technologies, cat. 15596026), following manufacturer's instructions for standard trizol-chloroform-ethanol extraction procedure. RNase-free glycogen was used as suggested by the protocol, to increase the yield of the RNA precipitation step. 2 μg of total RNA were reverse transcribed into cDNA by using Superscript III Reverse Transcriptase (Invitrogen, cat. 18080044) and a mixture of oligo(dT)18 primers (500 μg/mL); dNTP mix (10 mM); DTT (0.1 M); 5x First-Strand Buffer; RNaseOUT (40 units/μL).

For real-time PCR, SYBR Green Master mix (Bioline, cat. BIO-94020) was used. Primers are detailed in Supplementary Table 3. Technical replicates were carried out for all quantitative PCR. For mouse ESCs, human iPSCs and NPCs, Gapdh and GAPDH were used as endogenous control to normalise expression. The Ct mean of zebrafish *gapdh*, *eef1a1l1*, *tuba1b* and *b2m* was used as an endogenous housekeeping control for normalisation, due to the variability shown looking

at the expression levels of those genes. qPCR data were acquired with QuantStudio™ 6&7 Flex Software 1.0 and 1.3 version.

## RNA sequencing: Library Preparation

Total RNA was quantified using the Qubit 4.0 fluorimetric Assay (Thermo Fisher Scientific). Libraries were prepared from 125 ng of total RNA using the NEGEDIA Digital mRNA-seq research grade sequencing service (Negedia srl)[140] which included library preparation, quality assessment and sequencing on a NovaSeq 6000 sequencing system using a single-end, 100 cycles strategy (Illumina Inc.).

## Bioinformatics workflow

The raw data were analysed by Next Generation Diagnostics srl proprietary NEGEDIA Digital mRNA-seq pipeline, which involves a cleaning step by quality filtering and trimming, alignment to the reference genome and counting by gene[142,143]. Genes were sorted removing those that had a total number of counts below 5 in at least 3 samples out of 30. After applying this filter, we identified 11,851 expressed genes that were considered for further analyses.

Differential expression analysis was carried out in the R environment (v. 4.1.0) with Bioconductor (v. 3.14) exploiting the DESeq2 R package. (v. 1.34.0)[144] and edgeR (v. 3.36.0)[145]. DESeq2 performs the estimation of size factors, the estimation of dispersion for each gene and fits a negative binomial generalised linear model with two-tailed Wald statistics. Biological significance of DEGs between Q128 and Q15 (Fig. 1g), of DEGs between Q128_Mtf1 and Q128 (Supplementary Fig. 5b) and of genes rescued by *Mtf1* (Fig. 5c) was explored by Gene Ontology (GO) term enrichment analysis using Enrichr software (v. 3.0) and including the categories of Biological Processes (BP) (2021) and Molecular Function (MF) (2021).

To perform the enrichment of cell population proliferation (GO:0008283 and[146]) and apoptosis (R-HSA-109581 and WP1351) gene signatures in Q128 and Q15 cell lines overexpressing *Mtf1*, we used Gene Set Enrichment Analysis (GSEA) software (v 4.3.2)[147]. Pre-ranked gene set lists were generated based on $\log_2$ fold-change (FC) values as obtained by the differential expression analysis between Q128_Mtf1 vs Q128 and Q15_Mtf1 vs Q15.

Motif enrichment analysis was performed using the Motif Analysis tool from the Regulatory Genomics Toolbox suite (https://reg-gen.readthedocs.io). Mtf1 MRE was obtained from the Jaspar database (9th version, http://jaspar.genereg.net/)[148]. The command "rgt-motifanalysis matching" was used to search for binding sites on the promoter region of all the genes of interest; then, "rgt-motifanalysis enrichment" was used to perform a Fisher's exact test to evaluate if the proportion of binding sites in the gene set of interest is higher than expected by chance.

One representative biological replicate of RNA sequencing data and Mtf1 MRE were visualised as tracks in the Integrated Genomics Viewer (IGV v. 2.16.0) and shown in Fig. 5h.

Volcano plots and scatter plots were produced with $\log_2$ FC and $-\log_{10}$ p-value exploiting the ggscatter function from ggpubr R package (v. 0.4.0.5). Heatmaps were made using CPM values with the pheatmap function from pheatmap R package (v. 1.0.12).

## Alignment of Mtf1 orthologues

The *Mtf1* sequence alignment was performed using the Clustal Omega software (v1.2.4, https://www.ebi.ac.uk/Tools/msa/clustalo/)[149]. The identity between the sequences was calculated using the Sequence Manipulation Suite program (https://www.bioinformatics.org/sms2/ident_sim.html).

## Metal analysis

After a 48 hours conditioning treatment, $10^7$ Q15 and Q128 cells were collected, centrifuged in Eppendorf tubes, the supernatant was removed and the cell pellets were stored at −80 °C. Immediately before the analysis, cells were thawed and resuspended in concentrated $HNO_3$ (68%), 1 mL added to each sample. After complete dissolution, we added 2 mL of ultrapure water and fully mineralised the sample by means of microwave heating, using a high efficiency high pressure microwave reactor (Ultrawave, Milestone, Bergamo, Italy). The same procedure was applied to the culture medium, before and after the conditioning treatment. The calibration curves for the metals quantitation were obtained by preparing six standard solutions at different concentrations (0.005-0.010-0.025-0.050-0.100-0.200 ppm), making use of certified multi-element and single-element standards (Agilent). All the samples were then filtered through a 0.20 μm syringe filter. The metal analysis, with the atomisation and ionisation of the samples obtained in an Argon plasma, was performed by Inductively Coupled Plasma – Optical Emission Spectrometry (ICP-OES, mod 5110, Agilent).

## Generation of Zebrafish HD model

HD zebrafish were generated by microinjection in one-cell stage embryos of mRNA encoding the first exon of human *HTT* including 74Q or 16Q, fused in-frame to eGFP coding sequence. Q74eGFP and Q16eGFP were cloned into pCS2+ plasmid in order to allow for the in vitro transcription. pCS2_Mtf1 and pCS2_mCherry plasmids were also generated to obtain *Mtf1* and *mCherry* mRNAs used for injections in HD zebrafish embryos.

For RNA in vitro transcription, 2.5 μg of pCS2_Q74eGFP, pCS2_Q16eGFP, pCS2_Mtf1, and pCS2_mCherry were linearised by overnight digestion at 37 °C with HF-Not I (New England Biolabs, cat. R3189S). The digestion volume was then concentrated by the DNA Clean & Concentrator kit (Zymo Research, cat. D4003) and used for the capped transcription reaction (mMESSAGE mMACHINE™ SP6 Transcription Kit, Thermo Fisher Scientific, cat. AM1340) by SP6 RNA polymerase. After removing the DNA template by DNase treatment (Thermo Fisher Scientific, cat. AM2238) for 15 minutes at 37 °C, RNA was purified by Phenol-Chloroform extraction (as discussed in 'RNA isolation, reverse transcription and quantitative PCR' paragraph). RNA was quantified by Nanodrop ND-1000 and then diluted according to the need in a mix of 10% Danieau buffer [8 mM NaCl, 0.7 mM KCl, 0.4 mM $MgSO_4$, 0.6 mM $Ca(NO_3)_2$, 2.5 mM HEPES, pH 7.6], 10% Phenol Red (Merck, cat. 1072410025) and RNase-free water.

In order to select the injection dose that caused the highest rate of malformations with the lowest level of death, we injected increasing doses of Q74eGFP mRNA ranging from 150 to 1000 pg/embryo and phenotypically scored 24 hpf embryos. Once established the dose of 250 pg/embryo, under a light microscope, embryos were injected with in vitro transcribed mRNAs. Microinjected embryos were then transferred to fish water and incubated at 28 °C. Unfertilised eggs were recognised and discarded 4 hours post-microinjection. 24 hpf tadpoles were dechorionated using dedicated needles under a light microscope.

## Whole-mount stainings

Injected embryos were anaesthetised with tricaine and immobilised in 1.5% Methylcellulose or 2% low melting agarose and analysed using a Leica M165FC fluorescence microscope. Confocal zebrafish images were acquired with a Nikon C2 H600L confocal microscope.

For Acridine Orange hemi (zinc chloride) salt in vivo staining (Merck, cat. A6014), 24 hpf embryos were dechorionated, transferred into a 6-well plate and incubated in about 2 mL of Acridine Orange (20 μg/mL) per well in fish water for 15 minutes at 28 °C. The Acridine solution was then removed and embryos were washed three times with 1 mL of fish water. Before being observed on a glass slide by a fluorescence microscope, tadpoles were anaesthetised by Tricaine.

For the TUNEL assay, ApopTag Fluorescein In Situ Apoptosis Detection Kit (Merck, cat. S7110) and collagenase (Merck, cat. C9891)

were used. 7 embryos-30 hours post-microinjection per condition were placed in an Eppendorf, anaesthetised with Tricaine and fixed in 4% paraformaldehyde (PFA) at 4 °C overnight. Then, PFA was removed and samples were washed with PBS (3 times, 10 minutes each), while shaking. Embryos were dehydrated through a series of methanol solutions ranging from 10% to 100% and frozen at −20 °C overnight. Then, embryos were rehydrated with a series of 70-50-30% methanol solutions and washed with PBS with Tween-20 for 10 minutes, while shaking. After that, collagenase was applied for 8 minutes while shaking and the excess was washed away by PBS with Tween-20 washing steps (3 times, 5 minutes each). Samples were incubated for 1 hour in the equilibration buffer while shaking, then for 2 hours at 37 °C in working strength TdT. The reaction was stopped by washing twice the samples in the working strength Stop/Wash buffer. Next, there was a blocking step of 1 hour with PBS with Tween-20 while shaking, and then embryos were incubated overnight in a Working Strength anti-digoxigenin conjugate solution at 4 °C in the dark. The morning after, the antibody solution was removed, samples were washed with PBS (4 times, 10 minutes each) and analysed by a confocal microscope. Zebrafish larvae anterior structures were scanned in 70 stacks of 3.475 μm each, spanning their entire depth. We quantified the fractions of the fluorescent positive area over the total area (excluding the yolk region). For quantification analyses, all images were acquired with the same exposure parameters and processed using Fiji software (v2.9.0). Statistical analyses were carried out with Past (v.4.03) and Prism (v.9.5.0).

### AAV-PHP.eB vector injection, mouse phenotyping and tissue collection

AAV-PHP.eB viral particles were produced and titrated in Broccoli's lab as described previously[75]. This viral vector was modified to express under the control of the Ef-1α promoter the candidate gene *Mtf1* or either eGFP as a control. Vascular injection was performed in a restrainer that positioned the tail in a heated groove. The tail was swabbed with alcohol and then injected intravenously. WT and R6/2 mice were randomised in groups and injected in the tail vein at 4.2 weeks of age. Following injection, all mice were weighed twice a week. Phenotyping was carried out, blind to genotype and treatment, twice a week. The balance and the motor coordination were assessed by the Rotarod test and Horizontal Ladder Task. Total DNA was isolated from animal tissues (cortex and striatum) using the Qiagen DNeasy Blood and Tissue Kits (QIAGEN, cat. 9504).

### Animal husbandry

All Zebrafish experiments were carried out at the Fish Facility in the Department of Biology of the University of Padova. Zebrafish larvae were kept at most three days in Petri dishes with fish water (60 mg of Instant Ocean, cat. SS15-10, per litre of distilled water) at neutral pH at 28 °C, according to standard procedures (http://ZFIN.org).

Mouse colonies were established at IRCCS Neuromed. Breeding pairs of the R6/2 line of transgenic female mice [strain name: B6CBA-tgN (HDexon1) 62Gpb/1 J] with ~160 ± 10 CAG-repeat expansions were purchased from the Jackson Laboratories. Mice were housed under standard conditions (22 ± 1 °C, 60% relative humidity, 12 h light/dark schedule, 3–4 mice/cage, with free access to food and water). Male R6/2 mice (5-6 weeks of age) were crossed with female B6CBA WT mice (5-6 weeks of age) for colony maintenance; the resultant WT and R6/2 mice were used for all the experiments performed in this study. A complete list of mice used in this study, indicating age, sex, treatments and measurements, is reported in Supplementary Table 4. All experimental procedures were approved by the IRCCS Neuromed Animal Care Review Board ethics committee and by Istituto Superiore di Sanità (ISS permit number: 548/2022-PR) and were conducted according to the 2010/63/EU directive for animal experiments.

### Motor behaviour tests

All behaviour tests were carried out during the light phase of the light/dark cycle. Mice were tested before and after treatment at the indicated time points. Before training and testing, mice underwent a period of habituation to the testing room and equipment. All mice received training for two consecutive days on each instrument and task before performing motor behaviour measurements. Mice were tested at fixed speed (0.1 rcf) on a rotarod apparatus for 1 min. Each mouse was tested in three consecutive trials of 1 min each, with 1 min rest between trials. The time spent on the rotarod in each of the three trials was averaged to give the overall time for each mouse. In the horizontal ladder task, the mice spontaneously walked along a horizontal ladder with variable and irregular spacing between rungs. In each test session, the mouse performance was evaluated using an established footfall scoring system[150], which allows for qualitative and quantitative evaluation of forelimb and hindlimb placement on the ladder rungs. All motor tests were conducted by the same experimenter who was blinded to mouse genotype and experimental group throughout the entire course of the analysis.

### Clasping analysis

The clasping score is determined over 30 seconds. In particular, mice were suspended by their tails from a height of 50 cm and a limb-clasping response was defined as the withdrawal of any limb to the torso for more than 2 seconds. The following scores were used: 0 (absence of clasping), 0.5 (withdrawal of any single limb), 1 (withdrawal of any two limbs), 1.5 (withdrawal of any three limbs), 2 (withdrawal of all four limbs).

### Dihydroethidium (DHE)

WT and R6/2 mice were sacrificed by cervical dislocation. Brains were removed and trimmed by removing the olfactory bulbs and spinal cord. The remaining brain was processed and embedded in paraffin wax, 10 μm coronal sections were cut on an RM 2245 microtome (Leica Microsystems) and floated in a 40 °C water bath containing distilled water. Sections were transferred onto glass slides suitable for immunohistochemistry and let dry overnight at room temperature. Samples were deparaffinized in xylene for 30 minutes, transferred to 100% alcohol for 10 minutes and then once through 95%, 70% and 50% alcohol respectively for 10 minutes each, washed in PBS twice. In situ superoxide generation production was detected by fluorescence with DHE (Sigma-Aldrich, cat. D7008). Samples were incubated with DHE (2 μM) in a light-protected humidified chamber at 37 °C for 30 minutes. Slides were rinsed with PBS twice and observed at the microscope. For each staining, four mice per experimental group were used and three coronal sections for each animal were acquired with the Nikon ECLIPSE Ni microscope and analysed by NIS-Elements Image Software (v. 4.40, Nikon) and Fiji software 2.9.0.

### mHTT aggregates immunostaining

WT and R6/2 mice were sacrificed by cervical dislocation. Brains were removed and trimmed by removing the olfactory bulbs and spinal cord. The remaining brain was processed and embedded in paraffin wax, 10 μm coronal sections were cut on an RM 2245 microtome (Leica Microsystems) and floated in a 40 °C water bath containing distilled water. Sections were transferred onto glass slides suitable for immunohistochemistry and let dry overnight at room temperature. Samples were deparaffinized in xylene for 30 minutes, transferred to 100% alcohol for 10 minutes and then once through 95%, 70% and 50% alcohol respectively for 10 minutes each, washed in PBS twice. To unmask the antigenic epitope, antigen retrieval was performed using citrate buffer method (incubate with citrate buffer 10 mM, pH 6.0 at 95-100 °C for 15 minutes then allow slides to cool for 15 minutes). Slides were washed twice with PBS, permeabilized in TBS-Triton 0.1% for 10 minutes, then incubated in a humidified chamber at room

temperature with blocking buffer (Horse serum 10% in PBS) for 1 h. Blocking buffer was removed and slides were incubated in a humidified chamber at 4 °C overnight using a mouse anti-HTT antibody (clone EM48, for details see Supplementary Table 2). After washing three times with PBS, cells were incubated with secondary antibodies for 1 h in a humidified chamber at room temperature, protected from light. Nuclei were stained with DAPI (4′,6-diamidino-2-phenylindole; Sigma-Aldrich, cat. F6057). Four mice per experimental group were used and three coronal sections for each animal were acquired with the Nikon ECLIPSE Ni microscope and analysed by NIS-Elements Image Software (v. 4.40, Nikon) and Fiji software 2.9.0.

## Statistics and reproducibility

No statistical method was used to predetermine sample size, but our sample sizes are similar to those commonly used in our field of research. No data were excluded from the analyses. Data distribution was assumed to be normal but this was not formally tested. *P*-values for experiments involving repeated measures (Fig. 1c, Fig. 4f, Fig. 7b (left), 7c (left) and 7g, Fig. 8d,f, Supplementary Fig. 1e, Supplementary Fig. 4a,c, Supplementary Fig. 8c, Supplementary Fig. 9c) were calculated with Two-way Repeated Measure ANOVA with Bonferroni's correction. For experiments with cell lines, we randomly allocated a fraction of each cell population to different biological replicates. For the analysis of immunostaining and flow cytometry data, we analysed random fields or random fraction of cells. Other kinds of experiments were not randomised. Data collection and analysis were not performed blind to the conditions of the experiments, but data analyses have been performed with identical parameters and software. Analysis of mouse motor behaviours was performed in blind. Data representation and statistical analyses were performed using R software (v. 4.0.0 and v. 4.1.0) and PAST (v4.03), unless stated otherwise. All bars, error bars and box plots are defined in figure legends. The number of biological replicates and independent experiments, both >2, is indicated in figures legends. The statistical tests used are indicated in figure legends. All qPCR experiments were performed with three technical replicates. Key experimental results have been obtained by 2 independent operators.

## Reporting summary

Further information on research design is available in the Nature Portfolio Reporting Summary linked to this article.

## Data availability

The RNA sequencing data generated in this study have been deposited in the Gene Expression Omnibus (GEO) database under accession code GSE166567. All RNA-seq process data, used in Figs. 1f, g, 5b–d, Supplementary Fig. 5b–d and Supplementary Fig. 6d, e are reported in Supplementary Data 1, 2 and 3. Primers and oligonucleotide sequences are present in Supplementary Tables 1 and 3. A complete list of mice used in this study, indicating sex, treatments and measurements, is reported in Supplementary Table 4. Additional data that support the findings of this study, such as analysis pipelines and reagents are available from the corresponding authors upon reasonable request. All uncropped gels and numerical values are provided in the Source data file. Databases used in this study are HDNetDB (2017, http://hdnetdb.sysbiolab.eu), Cytoscape software (v3.8.2, http://www.cytoscape.org/), Enrichr database (v. 3.0, http://amp.pharm.mssm.edu/Enrichr), GSEA software (v. 4.3.2, http://software.broadinstitute.org/gsea/), Jaspar database (9th version, http://jaspar.genereg.net/), Clustal Omega software (v. 1.2.4, https://www.ebi.ac.uk/Tools/msa/clustalo/). Source data are provided with this paper.

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

## Acknowledgements

We would like to thank Sirio Dupont, Marco Montagner, Maria Pennuto, Stefano Ciaco, Rachele Ghirardo and the Martello lab for critical reading of the manuscript and discussions. We thank Elena Cattaneo's laboratory for plasmids and technical help. Neurogenetics Lab (Neuromed) is supported by Telethon Foundation Grants GJC21157-A and GGP20101, by the Italian Ministry of Health (Ricerca Corrente funding program) and Fondazione Neuromed. G.M.'s laboratory is supported by grants from the Giovanni Armenise–Harvard Foundation, Microsoft Research, the Telethon Foundation (TCP13013 and GJC21157-A) and an ERC Starting Grant (MetEpiStem).

## Author contributions

G.M. designed the study; G.M.F. performed ESC culture, generated mutant cell lines and performed the genome-wide screening. G.M.F., S.Am., A.M.G., P.R. and G.P. performed Western Blots. N.C., S.An., A.M.G. and E.M. performed zebrafish microinjections and molecular characterisations. M.A., L.D. and A.M.G. performed bioinformatic analyses on RNA-seq data. S.Am. and E.C. performed immunofluorescence experiments. A.M.G. and S.Am. performed iPSCs culture, NPCs differentiation and molecular characterisations. F.L.T. performed Mtf1 orthologues alignment. E.G., F.P. and M.Le identified PB integration sites by NGS. L.Z. and P.V. performed metal analysis. M.Lu, S.G. and V.B. generated AAV vectors and viral particles. G.P., L.C., A.D.P. performed tail-vein injection and phenotypic analyses in R6/2 mice. V.M. analysed the data and supervised the R6/2 studies. G.M. wrote the manuscript with inputs from all authors; G.M. supervised the study.

## Competing interests

G.M.F., A.M.G., M.Le. and G.M., files a patent application about the use of the suppressors identified in this study as therapeutic agents for polyQ diseases (Therapeutic factors for the treatment of PolyQ disease - Patent Application No. PCT/IB2023/051156 filed on 11 February 2022). Patent applicants are the institutions of Telethon Foundation, University of Padova and University of Wien. The remaining authors declare no competing interests.
