## [Peer Review File · Nature Communications]

A genome-wide screening in pluripotent cells identifies Mtf1 as a suppressor of mutant huntingtin toxicityREVIEWER COMMENTS

Reviewer #1 (Remarks to the Author):

In their study on suppressors of mutant huntingtin (mHTT) toxicity, the authors use a screening approach on pluripotent mouse ES cells to identify factors that could reduce mHTT toxicity. The authors generate ES cells that stably express an N-terminal fragment of WT (Q15) or mHTT (Q128) and observe increased stress and cell death in Q128 cells, as well as misregulation of genes that have previously been implicated in HD pathogenesis. The authors performed a gain-of-function screen for suppressors of mHTT toxicity by first treating them with further stressors and then selected for resistant mutants, identifying 5 candidate genes. Next, the authors performed a network analysis of the suppressor candidates, linking their effect to HD-relevant genes. They then chose Mtf1 for further studies, since it responded most efficiently to both stressors. Mtf1 was found to regulate HD-related genes and was shown to ameliorate HD-phenotypes in zebrafish and mouse models.

Overall, the paper is well-written and contains a number of interesting experiments with high-quality data and appropriate analysis, from screening to multiple in vitro and in vivo validations. The discovery of new regulators of HTT that ameliorate HD phenotypes is interesting. However, I also believe that the design of the study has flaws that can limit the interest to others in the community.

1. As the authors point out in the introduction, two important open problems in the HD field are why only specific types of neurons degenerate at a late stage, and the difficulty of translating findings in animal models to clinical trials. The authors write that "for this reason, we did not focus a priori on a specific cellular process, we chose instead to screen unbiasedly, on a genome-wide scale for factors able to reduce mutant HTT toxicity". While it is a good strategy to not focus on a specific cellular process and while the screening approach is generally interesting, it would have made more sense to screen or at least validate on cells that are closer to the ones affected in HD (i.e. human neurons or at least progenitors). Furthermore, both the usage of cell lines stably expressing Q15 or Q128 fragments and the addition of further stressors is quite artificial and will make it more difficult to determine whether the results are relevant for the treatment of HD. While the modeling of degenerative diseases remains a major challenge in the field, efforts that concentrate on getting closer to the phenotype rather than working on artificially generated phenotypes that may or may not have any relation to how the disease operates seem more promising.

2. As the authors write, "even though ameliorating cellular processes impaired in HD gave promising results in animal models, all clinical trials to date have not demonstrated efficacy." Therefore, although interesting in themselves, it is questionable whether the zebrafish experiments with their severe apoptosis phenotype during early development will be relevant to the pathophysiology of HD in humans. The mouse experiments are more relevant here, but given that the authors themselves mention the difficulty in translating findings from animal models to the clinic, which at the same time is a motivation for the study, they should at least comment on how they would approach this challenge in the future.

Minor comments:

- Figure legend 1b (line 854), incomplete sentence: "Q128 HTT protein expression resulted lower compared to Q15 HTT".
- (line 328) "This therapeutic approach has been thought in the framework..." consider rephrasing

Reviewer #2 (Remarks to the Author):

I have now read carefully the manuscript entitled "A genome-wide screening in pluripotent cells

identifies Mtf1 as a novel 1 suppressor of mutant huntingtin toxicity”, proposing Metal regulatory Transcription factor 1 (MTF1), a transcription factor controlling metal cellular homeostasis, as suppressor of mutant huntingtin-mediated toxicity.

Although the approach of finding genes able to prevent mutant huntingtin toxicity is not completely new, the technical method used (piggyBac (PB) mediated-insertional mutagenesis) is interesting and might reveal new hits with possible therapeutic potential.

However, the study suffers from major limitations mainly associated to the use of HD model systems which always over-express the N-terminal huntingtin fragment with (or without) expanded CAG tract. Without further validation in knock-in mouse embryonic stem cells, knock-in animal model (even with differ CAG alleles with lower CAG expansion and more closely mimicking the human HD mutation), patients’ derived cells lines, the authors findings have limited relevance, especially in consideration of further therapeutic development.

Little mechanistic insights are also proposed on how Mtf1 could (at least partially) rescue mutant huntingtin toxicity. This information will be crucial to understand whether Mtf1 over-expression could indeed be beneficial or detrimental for HD patients.

Some experimental controls using zebrafish models are missing, the data from the R6/2 mice treated with Mtf1 adenoviruses are somewhat preliminary and uniquely addressing behavioural changes. More molecular details would be required.

Major points:

1. All the models used (ES, zebrafish and mouse) are HD models over-expressing a fragment of N-terminal huntingtin. The authors should check whether results are reproducible in knock-in mouse embryonic stem cells (an isogenic panel of wild-type, heterozygous Htt CAG knock-in HdhQ20/7, HdhQ50/7, HdhQ91/7 and HdhQ111/7 mouse ES cell lines were described previously (White et al., 1997, Wheeler et al., 1999 and Jacobsen et al., 2011), knock-in animal model (Menalled et al., 2012, Fossale et al., 2011) and patients’ derived cells lines (HD iPSC consortium papers 2012 and 2017).
2. Not clear the data presented in Fig. 4e: is the over-expression of WT HTT fragment (Q15) causing a significant decrease in colony formation comparable to the one observed with Q128 fragment? Why is this? Moreover, this experiment is based on 2 technical replicates. Increasing the number of observations would be necessary.
3. Figure 5a. The over-expression of Mtf1 is inducing a partial (~30%) rescue of transcriptional alterations observed with overexpression of Q128 fragment. Would the combined over-expression of other target candidates (Kdm5b, Fbxo34 or others) improve this result? What are the genes ‘not-rescued’ by Mtf1? How is Mtf1 counteracting the transcriptional dysregulation elicited by HTT Q128 fragment over-expression? More mechanistic insights would be required.
4. Why pointing all the attention only on Mtf1? Not completely clear the rational for this choice.
5. Figure 6, zebrafish experiments: what is the percentage of healthy, mildly and strongly affected embryos when only Mtf1 is injected? How are the authors explaining the huge variability in embryo phenotypes obtained after HTT-Q74 administration?
6. Figure 7, mouse R6/2 behavioural phenotypes. What is happening at the morphological/molecular levels when the mice receive Mtf1 viral particles? Was a decrease in apoptosis/ROS production observed?
7. The discussion session is too short. Many literature data, possibly interesting to frame this study, are not cited or discussed (i.e. iron dysregulation in HD)

Minor points:

1. In figure 1b, why do they see a lower expression of the Q128 construct compared with the Q15 one?
2. Treated mice are tested at 11 week for their motor impairment. What about later stages? Would they need another shot of treatment, or how long does the effect last?
3. Lines 230 and 232: refers to Supplementary Fig. 5a/5b, while instead it should be referring to Supplementary Fig. 6a/6b
4. Lettering, text size of the figure is not always comparable across the manuscript.

Reviewer #3 (Remarks to the Author):

This paper reports a screening strategy that has identified a novel suppressor of polyglutamine toxicity. This is interesting for me as the authors have used a strategy that I would not have predicted would be successful if asked advice at the outset – but it has worked so this is interesting for me.

The study has been well conducted and the paper flow is clear – the English could do with some editing but this is stylistic and the flow is logical and the data are presented well.

The strength of the paper is the successful screening strategy in cells that then is translated into zebrafish and mouse models of Huntington disease.

The weakness is that the authors do not provide or really seek to explore a mechanism by which Mtf1 overexpression suppresses polyglutamine toxicity. (I use the term polyglutamine toxicity as this exon 1 of huntingtin really just has the polyglutamine tract with a little extra and the toxicity is likely primarily due to the polyglutamines.) I would suggest that the authors consider studying the effect of the Mtf1 on the mutant huntingtin abundance in both soluble forms and aggregates in the cell and mouse models and looking to see if this provides any clues about how Mtf1 is protective. If the abundance is changed then the authors could look at mechanisms, like the ubiquitin-proteasome pathway or autophagy. Some mechanistic understanding would make a big difference to the impact of the paper. (In Fig 4c the levels of “soluble huntingtin look as they may be lower with the Mtf1 compared to the controls.) Also, the authors may want to consider if Mtf1 overexpression protects against cell death insults like staurosporine, hydrogen peroxide, rotenone etc., to assess whether the mechanism is associated with generic protection against cell death compared to a process that is more specific.

Reviewer #4 (Remarks to the Author):

This manuscript entitled “A genome-wide screen in pluripotent cells identifies Mtf1 as a novel suppressor of mutant huntingtin toxicity” uses a clever and fairly stringent survival screen for suppressors of toxicity in a new Q128 Huntington disease mouse ES cell model. The screen itself utilized an MSCV system that upon random integration upregulates neighborhood genes. This revealed possible potent suppressors of HTT N-terminal Q128 toxicity in the presence of other stressors.

The major strength of the work is its use of an unbiased, genome-wide approach followed by additional efforts to validate a top hit in both zebrafish and R6/2 mice, the latter even using AAV/Php.Eb to achieve CNS expression of Mtf1 through venous delivery. This provides a nice proof-of-concept pipeline for a forward genetic screen in culture to efficient in vivo validation that others could follow.

One limitation of the study is that it does not examine mutant HTT aggregation (either biochemically or histologically), an important aspect of Huntington disease pathogenesis, and how it might be affected by Mtf1 expression. In addition, there is also only limited evidence showing that protective effects of Mtf1 are specific to mutant HTT and this could be characterized further in the non-mutant HTT settings. The study is otherwise technically sound and their overall conclusions appear to be well supported by their experiments. The manuscript is written well and easy to follow with several helpful diagrams provided along the way. Here are some suggestions that could help address the above limitations along with other minor issues:

1. The study should strongly consider characterizing mutant HTT aggregation phenotypes in these different systems to determine if Mtf1 has effects on this important aspect of HD pathogenesis. Reporting such findings would significantly strengthen the study towards understanding why Mtf1 is protective and if it's acting upstream or downstream of misfolded HTT accumulation.
 - a. Figure 1b shows lower Q128 monomeric levels, suggesting there may be increased aggregation in the ES cells. Figure 4c does not demonstrate any change in the monomer levels, so perhaps the effects of these suppressors could be independent of the propensity of mutant HTT to aggregate. Nonetheless, evaluation of the Q128 cell model by filter trap, evaluation of the stack of the western blot, and/or by immunocytochemistry would be worth reporting. ES cells may not show much or any aggregation phenotypes.
 - b. Similarly, there appears to be punctate staining in the Q74 condition in zebrafish (Figure 6b) that is reduced by Mtf1. Are these aggregated proteins? A closer examination of this potential phenotype would be helpful.
 - c. Evaluation of the R6/2 mouse brains by western blotting and immunohistochemistry for mutant HTT aggregation and inclusion formation, respectively, would again help examine if Mtf1 affects mutant HTT aggregation. Moreover, it would also further reveal the extent to which this AAV delivery drives Mtf1 expression in neurons versus non-neuronal cell types and distribution (e.g. cortex versus striatum). Finally, IHC would help determine if there is rescue of neurodegeneration phenotypes (e.g. by DARPP32 staining of the striatum) beyond only the behavioral phenotypes reported.
2. A relatively underexplored control in this study is the effects of the putative suppressors on Q15-ES cells. This is briefly addressed in figure 5e and supplemental figure 5, which show no significant effects of Mtf1 expression on Q15 or the ES parental line. It is somewhat difficult with only these pieces of data alone to confidently know if Mtf1 effects are mutant HTT-specific versus protective against more generalized toxicity, especially given the noisiness of the data in figure 5e.
 - a. In figure 4e, can Mtf1 also rescue toxicity in Q15_EV with MG132/tamoxifen? Or is this effect only seen in Q128? This can be similarly asked for another putative suppressor. If so, this might suggest that Mtf1 is generally protective against general stress (as alluded to by the authors in discussion).
 - b. Consider moving both Figure 5e and supplemental figure 5 to figure 4, where one would first ask if the effects of the suppressors are Q128-specific.
3. There appears to be a robust effect on Cresyl violet intensity with Kdm5b in figure 4e, but no significant effect on the number of cells in Figure 4d - why? Perhaps reporting the results of n=3 experiments in 4e could help resolve the discrepancy (rather than showing a single experiment with technical duplicate in 4e). Relatedly, the comparisons in figure 4e are difficult to follow. Consider breaking apart Figure 4e with separate panels for untreated, MG132, and Tamoxifen treated conditions, so Q15, Q128, and Q128_Mtf1 etc can be compared directly side-by-side.
4. This manuscript indicates Mtf1 is a transcription factor that translocates to the nucleus during stress - can this be assessed by immunocytochemistry in the Q128 versus Q15 condition? Also can the authors comment on Mtf1 conservations across species, especially since murine Mtf1 appears to have an effect in zebrafish? Is there an ortholog in zebrafish? This could be included in the discussion.
5. In figure 5, the authors show rescue of mutant HTT-induced transcriptional changes by Mtf1 overexpression. However, what are the total number of DEGs in (Q128_MTF1 vs Q15_EV) and (Q128_EV vs Q15_EV)? Is the rescue of DEGs in Q128_Mtf1 also associated with fewer total DEGs? Or does Mtf1 overexpression drive a separate robust transcriptional program with even more DEGs? A small supplementary bar graph of the total DEGs in these conditions could show this.
 - b. Similarly, are RNA-seq data available for Mtf1 expression by itself or in the Q15 condition? Would that lead to a similar gene enrichment analysis in figure 5d? It would be helpful to know if there is any overlap with the DEGs of the Q128 line (relative to Q15), especially if DEGs are moving in the opposing direction.

Other minor comments:

1. Considering moving supplemental figure 4 into the main figure 4 (alongside 4c) if space allows.
2. Consider adding detail in the abstract for the type of model used for the screen. It currently only mentions "pluripotent stem cells," which could be misinterpreted as patient-derived iPSCs.
3. In Figure 7c,d,e, consider using red lines for both R6/2 mouse conditions so WT AAV-Mtf1 and R6/2

AAV-GFP black dashed lines do not get confused with each other.

5. For figure 7c,d,e top-half, consider an ANOVA statistical analysis for the longitudinal behavioral study to highlight significant differences.

6. The p value is quite low for Figure 4d, Q128_Fbxo34, despite nearly overlapping error bars - can the authors confirm this is the correct p value?

7. Figure 3c, it is not clear to me NGS nodes are "pink" as described and would consider using a different color. The HD interacting gene nodes look more pink.

8. Page 2, line 17 italicize *in vitro*

9. Page 3, line 26: Define polyQ as polyglutamine for first use. In line 29, can then say polyQ-encoding.

10. Page 3, line 39-41 "For this reason.. reduce mutant toxicity": separate into two sentences.

11. Page 7, line 109 - "...44 mutant clonal lines *that* emerged..."

POINT-BY-POINT RESPONSE TO REVIEWER COMMENTS

Reviewer #1 (Remarks to the Author):

In their study on suppressors of mutant huntingtin (mHTT) toxicity, the authors use a screening approach on pluripotent mouse ES cells to identify factors that could reduce mHTT toxicity. The authors generate ES cells that stably express an N-terminal fragment of WT (Q15) or mHTT (Q128) and observe increased stress and cell death in Q128 cells, as well as misregulation of genes that have previously been implicated in HD pathogenesis. The authors performed a gain-of-function screen for suppressors of mHTT toxicity by first treating them with further stressors and then selected for resistant mutants, identifying 5 candidate genes. Next, the authors performed a network analysis of the suppressor candidates, linking their effect to HD-relevant genes. They then chose Mtf1 for further studies, since it responded most efficiently to both stressors. Mtf1 was found to regulate HD-related genes and was shown to ameliorate HD-phenotypes in zebrafish and mouse models.

Overall, the paper is well-written and contains a number of interesting experiments with high-quality data and appropriate analysis, from screening to multiple in vitro and in vivo validations. The discovery of new regulators of HTT that ameliorate HD phenotypes is interesting. However, I also believe that the design of the study has flaws that can limit the interest to others in the community.

1. As the authors point out in the introduction, two important open problems in the HD field are why only specific types of neurons degenerate at a late stage, and the difficulty of translating findings in animal models to clinical trials. The authors write that "for this reason, we did not focus a priori on a specific cellular process, we chose instead to screen unbiasedly, on a genome-wide scale for factors able to reduce mutant HTT toxicity". While it is a good strategy to not focus on a specific cellular process and while the screening approach is generally interesting, it would have made more sense to screen or at least validate on cells that are closer to the ones affected in HD (i.e. human neurons or at least progenitors). Furthermore, both the usage of cell lines stably expressing Q15 or Q128 fragments and the addition of further stressors is quite artificial and will make it more difficult to determine whether the results are relevant for the treatment of HD. While the modeling of degenerative diseases remains a major challenge in the field, efforts that concentrate on getting closer to the phenotype rather than working on artificially generated phenotypes that may or may not have any relation to how the disease operates seem more promising.

We thank the Reviewer for suggesting to validate our findings in human cells closer to those affected in HD. We used neural precursor cells (NPCs), for which mutant HTT-specific phenotypes have been reported¹⁻³. Importantly, these NPCs have been derived from iPSCs obtained from HD patients and healthy controls, thus the endogenous HTT gene bears the mutation and there is no exogenous expression of fragments.

After confirming loss of pluripotency and expression of neural progenitor markers SOX1, NESTIN and PAX6 by immunostaining and qPCR in NPCs (Fig. 8c and Supplementary Fig. 9a-b), we measured the levels of ROS production, which we found

increased in presence of mHTT also observed in mouse ESCs (Fig. 1e) and in the brain of R6/2 mice (Fig. 7f-g). NPCs derived from HD patients showed increased ROS production (Fig. 8e) and cell death (Fig. 8f), as reported in literature^{1,3-6}.

We then performed a Mtf1 overexpression in NPCs and observed a reduction in ROS production and cell death (Fig. 8e-f), as also observed in mouse ESCs (Fig. 5f-g).

In conclusion we validated Mtf1 in a human neuronal model (NPCs) derived from iPSCs thus confirming its protective effect in a cell type that is closer to the ones affected in HD.

2. As the authors write, "even though ameliorating cellular processes impaired in HD gave promising results in animal models, all clinical trials to date have not demonstrated efficacy." Therefore, although interesting in themselves, it is questionable whether the zebrafish experiments with their severe apoptosis phenotype during early development will be relevant to the pathophysiology of HD in humans. The mouse experiments are more relevant here, but given that the authors themselves mention the difficulty in translating findings from animal models to the clinic, which at the same time is a motivation for the study, they should at least comment on how they would approach this challenge in the future.

We completely agree on the difficulties in translating our findings from animal models to the clinic. We have now discussed this point (Page 22, Line 468-472). Our strategy is two fold: on the one hand we believe that drugs activating the expression of Mtf1 should have a protective effect in animal models and in HD patients; on the other hand, a gene therapy approach based on AAV-mediated delivery of Mtf1, as performed in R6/2 mice could be technically challenging (e.g. stereotaxis would be needed to transduce the striatum), but potentially effective.

Minor comments:

- Figure legend 1b (line 854), incomplete sentence: "Q128 HTT protein expression resulted lower compared to Q15 HTT".
- (line 328) "This therapeutic approach has been thought in the framework..." consider rephrasing

We rewrote the sentences indicated by the Reviewer (Line 460 and 1242).

Reviewer #2 (Remarks to the Author):

I have now read carefully the manuscript entitled “A genome-wide screening in pluripotent cells identifies Mtf1 as a novel 1 suppressor of mutant huntingtin toxicity”, proposing Metal regulatory Transcription factor 1 (MTF1), a transcription factor controlling metal cellular homeostasis, as suppressor of mutant huntingtin-mediated toxicity.

Although the approach of finding genes able to prevent mutant huntingtin toxicity is not completely new, the technical method used (piggyBac (PB) mediated-insertional mutagenesis) is interesting and might reveal new hits with possible therapeutic potential. However, the study suffers from major limitations mainly associated to the use of HD model systems which always over-express the N-terminal huntingtin fragment with (or without) expanded CAG tract. Without further validation in knock-in mouse embryonic stem cells, knock-in animal model (even with differ CAG alleles with lower CAG expansion and more closely mimicking the human HD mutation), patients' derived cells lines, the authors findings have limited relevance, especially in consideration of further therapeutic development. Little mechanistic insights are also proposed on how Mtf1 could (at least partially) rescue mutant huntingtin toxicity. This information will be crucial to understand whether Mtf1 over-expression could indeed be beneficial or detrimental for HD patients.

Some experimental controls using zebrafish models are missing, the data from the R6/2 mice treated with Mtf1 adenoviruses are somewhat preliminary and uniquely addressing behavioural changes. More molecular details would be required.

Major points:

1. All the models used (ES, zebrafish and mouse) are HD models over-expressing a fragment of N-terminal huntingtin. The authors should check whether results are reproducible in knock-in mouse embryonic stem cells (an isogenic panel of wild-type, heterozygous Htt CAG knock-in HdhQ20/7, HdhQ50/7, HdhQ91/7 and HdhQ111/7 mouse ES cell lines were described previously (White et al., 1997, Wheeler et al., 1999 and Jacobsen et al., 2011), knock-in animal model (Menalled et al., 2012, Fossale et al., 2011) and patients' derived cells lines (HD iPSC consortium papers 2012 and 2017).

We thank the reviewer for raising this important issue, concerning the use of knock-in models, rather than systems in which a fragment is overexpressed. In order to address also a similar point raised by Reviewer #1, we decided to use human neural precursor cells obtained from iPSCs derived from fibroblasts of HD patients and healthy controls. In this way we rule out that the effects we observed are due to overexpression artefacts, and we also validate our findings in human cells.

After confirming loss of pluripotency and expression of neural progenitor markers SOX1, NESTIN and PAX6 by immunostaining and qPCR in NPCs (Fig. 8c and Supplementary Fig. 9a-b), we measured the levels of ROS production, which we found altered also in mouse ESCs (Fig. 1e) and in the brain of R6/2 mice (Fig. 7f-g). As also reported by other studies^{2-4,6}, NPCs derived from HD patients showed increased ROS production (Fig. 8e).

We then performed a Mtf1 overexpression in NPCs and observed a reduction in ROS production (Fig. 8e), as also observed in mouse ESCs (Fig. 5f).

Importantly, we observed that NPCs derived from HD-patients were more susceptible to oxidative stress caused by Rotenone (Fig. 8f), as reported by other studies^{2-4,6}.

Indeed, we observed an increase in the number of AnnexinV-positive, apoptotic cells in Q109 NPCs, which was reduced by Mtf1 overexpression (Fig. 8f). We conclude that our results are reproducible also in a model in which mutant HTT is not overexpressed, but rather expressed from the endogenous locus.

We are aware of the fact that the Reviewer suggested also to perform additional experiments in knock-in mouse models or knock-in ESCs, however those experiments would require at least one more year of work and optimisation, but they would not inform us about MTF1 function more than the experiments we performed with human NPCs.

We are considering using the knock-in mouse model in the future to develop novel methods to activate MTF1 for a long time, but this will be a new project to develop over the next 3 years.

2. Not clear the data presented in Fig. 4e: is the over-expression of WT HTT fragment (Q15) causing a significant decrease in colony formation comparable to the one observed with Q128 fragment? Why is this? Moreover, this experiment is based on 2 technical replicates. Increasing the number of observations would be necessary.

First of all we should apologise for a mistake in the labelling of the samples during figure preparation. The “Q15” sample was, in fact, a Q128 sample expressing an empty vector. The correct identity of samples was indicated in the source data file provided at the time of the first submission.

However, we followed the suggestions of both Reviewer #2 and #3, by presenting data from at least 4 independent experiments, to improve the robustness of our conclusions. We also presented results for untreated (Fig. 4e-f and Supplementary Fig. 4a) and stressors-treated (Fig. 4g) samples in separate panels, to make data presentation more clear.

We further confirmed that Mtf1 and Kdm2b significantly increased the number of Q128 cells. Conversely, in the presence of stressors, only Mtf1 was able to increase the number of Q128 cells significantly for both stressors.

3. Figure 5a. The over-expression of Mtf1 is inducing a partial (~30%) rescue of transcriptional alterations observed with overexpression of Q128 fragment. Would the combined over-expression of other target candidates (Kdm5b, Fbxo34 or others) improve this result?

In the absence of stressors, Mtf1 and Kdm2b were both able to significantly increase the number of Q128 cells (Fig. 4e-f). Thus, we chose those 2 candidates for the combined over-expression suggested by the Reviewer.

We generated a construct containing both candidates, obtained ESCs stably expressing either (Q128_Mtf1 and Q128_Kdm2b) or both candidates (Q128_Mtf1+Kdm2b). We verified that the expression of transgenes was comparable in all lines generated (Supplementary Fig. 6c). This highly controlled experimental setup allows us to directly compare the transcriptional effects of 2 candidates.

As both MTF1 and KDM2B are transcriptional regulators, we measured their effects by transcriptomic analysis. MTF1 rescues a fraction (36.8%) of DEGs regulated by mHTT. KDM2B rescued a similar fraction (39.7%), while their combined expression did not increase the numbers of DEGs (37.6%). So the combined over-expression of MTF1 and KDM2B was not synergistic, it did not improve the rescue effect. An independent analysis on the magnitude of either gene up- or down-regulation by mHTT, confirmed the lack of synergy between the two suppressors (Supplementary Fig. 6d-e).

We conclude that MTF1 and KDM2B have similar effects on the transcriptome of Q128 cells and that they do not act synergistically in this context. We speculate that the combination of two suppressors working on similar processes (i.e. transcription) is less likely to display synergy. In contrast, combining two candidates acting on unrelated processes (e.g. a transcription factor and an ubiquitin ligase) are more likely to show synergistic effects. We will investigate this aspect more systematically in the future.

What are the genes 'not-rescued' by Mtf1?

To address this point, we first performed a more comprehensive characterisation of the genes and processes regulated by mHTT (Fig. 1g). We both increased the number of independent samples (>6) analysed and used multiple databases for gene set enrichment analyses. As a consequence we are more confident about the results we obtained.

We then asked which processes were rescued by MTF1. MTF1 rescued glutathione, neutral amino acid, steroid and iron metabolism, while other processes, such as histone modifications, IL-12 binding, superoxide and glycogen metabolic processes, were not affected by MTF1 expression (Fig. 5c).

How is Mtf1 counteracting the transcriptional dysregulation elicited by HTT Q128 fragment over-expression? More mechanistic insights would be required.

We agree on the importance of dissecting the mechanism of action of Mtf1 in Q128 cells. For this reason we started from the analysis of possible upstream activators of Mtf1.

An increased ROS production, which we observed in Q128 cells, is known to activate MTF1⁷.

Given that accumulation of metals, such as Cadmium, Zinc or Iron, could activate Mtf1⁸ we asked whether the presence of mutant HTT might affect metal homeostasis in the ESCs. We thus cultured Q15 and Q128 cells for 48 hours and performed Inductively Coupled Plasma Optical Emission Spectrometry (ICP-OES), an highly sensitive method for detection of elements, to measure the concentration of a panel of metals, both in the cells and in the media. We observed a strong increase in the concentration of Cadmium and Selenium in Q128 cells relative to Q15 cells (Fig. 5g). As an independent confirmation, we also measured the concentration of metals in media in which the cells were cultured, as well as in fresh medium, and found the Cadmium

concentration was significantly lower in medium conditioned by Q128 cells (Fig. 5g). These results indicate that Q128 cells accumulate Cadmium from the medium. Of note, a similar increase in intracellular levels of Cadmium has been observed also in striatal cells⁹⁻¹¹, which was linked to increased ROS production and cytotoxicity.

In Q128 cells we observed increased ROS production and accumulation of Cadmium (Fig. 1e and 5g), two stimuli that could activate MTF1. However, we measured by qPCR and RNAseq the expression levels of *Mt1* and *Mt2*, as a proxy of *Mtf1* activity (Supplementary Fig. 6a and Supplementary Table 1) and found no significant differences in Q128 cells, indicating that endogenous Mtf1 is not activated in response to cytotoxic effects caused by mHTT. We performed immunofluorescence and endogenous MTF1 was barely detectable in Q128 cells, which explains why over-expression of MTF1 is needed to counteract the detrimental effects elicited by mHTT (Supplementary Fig. 6b).

We then analysed the nuclear or cytoplasmic localisation of MTF1 in Q128_Mtf1 cells. Mtf1 should shuttle from the cytoplasm to the nucleus in response to accumulation of metals or ROS. However several transcription factors can also localise to other organelles (e.g. Stat3 in the mitochondria¹²) or bind RNAs in the cytoplasm rather than the genomic DNA^{13,14}. Quantitative immunofluorescence revealed that Mtf1 is mostly localised in the nucleus of Q128_Mtf1 cells (Fig. 5a). Interestingly, the nuclear localisation is more pronounced in Q128 than in Q15 cells (Reviewer Fig. 1). However, a more thorough investigation of how mHTT affects MTF1 localisation will be ground for future investigations.

Reviewer Figure 1

Immunofluorescence for MTF1 in Q15_Mtf1 and Q128_Mtf1 cells. Nuclei were stained with DAPI. Scale bar, 30 μ m. Please note in the merge that nuclear MTF1 signal is more pronounced in Q128_Mtf1 cells.

We conclude that Mtf1 is mostly nuclearly localised in Q128, suggesting that it might counteract the transcriptional dysregulation elicited by mutant HTT by directly regulating gene expression.

Metal response elements^{15,16} (MRE) are DNA sequences found on the promoters of genes regulated by the concentration of metals in a MTF1-dependent manner. We analysed the promoter of genes regulated by MTF1 and found that a significant fraction contained MREs (11.83% *p*-value 1.54e-4 Fisher's exact test), consistent with

a direct transcriptional regulation of Mtf1 on target genes. Importantly, 12.8% of genes misregulated by mHTT and rescued by MTF1, contained MREs.

We conclude that in Q128 cells the endogenous Mtf1 pathway is not active. Exogenous MTF1 localises mainly to the nucleus and counteracts the transcriptional dysregulation elicited by mHTT, via transcriptional regulation of genes containing MREs on their promoter.

4. Why pointing all the attention only on Mtf1? Not completely clear the rational for this choice.

Mtf1 displayed a stronger effect in Q128 cells and was also more effective in the presence of both stressors. This is more clear in the revised version of the manuscript (Fig. 4e-g). For these reasons it was used for further characterisation and following *in vivo* experiments. We will study in more detail other candidates in the future.

5. Figure 6, zebrafish experiments: what is the percentage of healthy, mildly and strongly affected embryos when only Mtf1 is injected? How are the authors explaining the huge variability in embryo phenotypes obtained after HTT-Q74 administration?

We apologise for omitting this relevant piece of information. Mtf1 expression by itself, even at high doses (i.g. 75ng/embryo) did not alter development, as we obtained 27 out of 27 healthy embryos. Please see Page 15, Line 304.

The variability in the phenotypes observed is expected from microinjection experiments. We should stress that each datapoint in Figure 6c represents 1 out of 8 independent experiments, performed by 3 different operators, for a total of >1500 embryos. Differences in the quality of embryos obtained, the skills of the operator and other variables contribute to the variability observed. However, despite this variability, Mtf1 effect is highly significant and reproducible.

6. Figure 7, mouse R6/2 behavioural phenotypes. What is happening at the morphological/molecular levels when the mice receive Mtf1 viral particles? Was a decrease in apoptosis/ROS production observed?

We measured the levels of ROS production and observed an increase in R6/2 mice, relative to wild-type littermates (Fig. 7f-g). Mtf1 delivery brought back to endogenous levels ROS production.

As an additional HD-related readout, we measured the formation of protein aggregates in the striatum, both via Western Blot and Immunostaining. Mtf1 delivery reduced the formation of aggregates (Fig. 7h-k).

Finally, we used brain weight as an indirect measure of brain atrophy, which was significantly reduced in R6/2 mice transduced with AAV-GFP, relative to control littermates (Fig. 7e). Crucially, R6/2 mice transduced with AAV-Mtf1 showed only a mild reduction, which did not reach significance, indicating a partial protective effect of Mtf1.

7. The discussion section is too short. Many literature data, possibly interesting to frame this study, are not cited or discussed (i.e. iron dysregulation in HD)

We apologise for that. We have significantly expanded the Discussion, citing key works about the role of metals in neurodegenerative diseases.

Minor points:

1. In figure 1b, why do they see a lower expression of the Q128 construct compared with the Q15 one?

The Reviewer raised an interesting question, as we also noticed this difference in protein levels, despite the similar mRNA levels. Similar reductions in mHTT protein levels, relative to the wild-type protein, have been observed in knock-in models, both *in vitro* in ES cells^{17,18} and in striatal cells¹⁹⁻²¹, but also *in vivo* (i.e. CAG-140 knock-in mice²²) as well as in patients samples²³.

Thus, our results are in line with previous studies. Lower levels of mHTT could be due to a lower affinity of anti-HTT for the mutant form, as suggested in Jacobsen and colleagues¹⁷. However, formation of insoluble aggregates or differences in translatability of a mRNA containing an elevated number of CAG repeats could also explain the reduced levels of mHTT.

2. Treated mice are tested at 11 week for their motor impairment. What about later stages? Would they need another shot of treatment, or how long does the effect last?

We had to stop our experiments at 11 weeks for ethical reasons, as the R6/2 mice treated with AAV-GFP displayed strong symptoms and were suffering. We agree that in the future we should perform new studies to measure how long the effect lasts, potentially using mouse knock-in models, in which symptoms arise more gradually, but we think such experiments go beyond the scope of the current manuscript.

3. Lines 230 and 232: refers to Supplementary Fig. 5a/5b, while instead it should be referring to Supplementary Fig. 6a/6b

We apologise for the mistake, we carefully checked all references to figures.

4. Lettering, text size of the figure is not always comparable across the manuscript.

We made sure that text size is consistent across all figures.

Reviewer #3 (Remarks to the Author):

This paper reports a screening strategy that has identified a novel suppressor of polyglutamine toxicity. This is interesting for me as the authors have used a strategy that I would not have predicted would be successful if asked advice at the outset – but it has worked so this is interesting for me.

The study has been well conducted and the paper flow is clear – the English could do with some editing but this is stylistic and the flow is logical and the data are presented well. The strength of the paper is the successful screening strategy in cells that then is translated into zebrafish and mouse models of Huntington disease.

The weakness is that the authors do not provide or really seek to explore a mechanism by which Mtf1 overexpression suppresses polyglutamine toxicity. (I use the term **polyglutamine toxicity** as this exon 1 of huntingtin really just has the polyglutamine tract with a little extra and the toxicity is likely primarily due to the polyglutamines.) I would suggest that the authors consider studying the effect of the Mtf1 on the mutant huntingtin abundance in both soluble forms and aggregates in the cell and mouse models and looking to see if this provides any clues about how Mtf1 is protective. If the abundance is changed then the authors could look at mechanisms, like the ubiquitin-proteasome pathway or autophagy. Some mechanistic understanding would make a big difference to the impact of the paper. (In Fig. 4c the levels of “soluble huntingtin look as they may be lower with the Mtf1 compared to the controls.) Also, the authors may want to consider if Mtf1 overexpression protects against cell death insults like staurosporine, hydrogen peroxide, rotenone etc., to assess whether the mechanism is associated with generic protection against cell death compared to a process that is more specific.

We thank the Reviewer for the constructive suggestions, which helped us in clarifying how Mtf1 suppressed polyglutamine toxicity.

As suggested, we looked at the levels of mutant HTT protein and aggregates formation, both *in vitro* and *in vivo*.

First, in mouse ESCs we measured the soluble protein levels of mutant HTT, both in the original clone expressing Mtf1 (MG18, Fig. 2g) and in the overexpressing lines generated (Q128_Mtf1, Fig. 4d) and in both cases observed a mild reduction of 17% and 14%, respectively, relative to controls. We conclude that there is no major effect on the levels of soluble HTT upon Mtf1 expression.

Concerning the formation of aggregates, we could not detect aggregates in ESCs, both by immunostaining and Western blot for insoluble proteins (Supplementary Fig. 1b). Previous studies based on knock-in mouse ESCs also did not report aggregates formation in pluripotent cells^{17,18,24–26}. Also in human pluripotent stem cells, aggregates have been detected only after neural differentiation upon overexpression of mHTT²⁷, indicating that protein aggregation is hard to detect in highly proliferative pluripotent cells.

Next, we analysed the protein aggregates in R6/2 mice and detected robust formation of aggregates with both techniques (Fig. 7h-k), which were reduced by Mtf1 delivery. We conclude that Mtf1 reduces mutant HTT protein aggregation.

Interestingly, several studies linked protein aggregation and oxidative stress in neurodegenerative diseases^{28,29}. We investigated ROS production in R6/2 mice and observed increased production, relative to wild-type littermates, which was normalised by Mtf1 delivery (Fig. 7f-g), potentially linking oxidative stress and protein aggregation in R6/2 mice.

We then looked at two markers of autophagy, LC3 and ATG7, and found that both were induced by Mtf1 only in the striatum of R6/2 mice (Reviewer Fig. 2). However, these results about autophagy are preliminary and we intend to explore further the role of autophagy in the future.

Reviewer Figure 2

a,b Representative western blotting of LC3 protein in striatal (**a**) and cortical (**b**) tissues from GFP- and Mtf1-treated R6/2 mice at 8 weeks of age. N=5 mice for each experimental group. p-values were calculated with Two-tailed Unpaired T-test.

c,d Representative cropped western blotting of ATG7 protein in striatal (**c**) and cortical (**d**) tissues from GFP- and Mtf1-treated R6/2 mice at 8 weeks of age. N=5 mice for each experimental group. p-values were calculated with Two-tailed Unpaired T-test.

A second crucial suggestion concerns the protective effect of Mtf1, asking whether “the mechanism is associated with generic protection against cell death compared to a process that is more specific”.

We thus generated control ESCs expressing an empty vector or a vector encoding the mCherry or a vector encoding for Mtf1. After verifying Mtf1 was overexpressed and its direct targets Mt1 and Mt2 were strongly induced (Supplementary Fig. 4b), we exposed cells to 7 different compounds inducing cell death. Notably, Mtf1 did not show any protective effect (Supplementary Fig. 4d).

We also performed transcriptomic analysis on both Q15 and Q128 cells expressing Mtf1 and failed to detect significant regulation of genes involved in proliferation or apoptosis (Supplementary Fig. 5b-c), further indicating that MTF1 does confer a generic resistance to cell death.

We conclude that in healthy cells, which do not express mHTT, Mtf1 does not exert any protective effect against cellular stress, while it shows protective effects in several models (mESCs, zebrafish, human NPCs and mouse models) in which mHTT causes cellular stress and death.

We will investigate in the future whether MTF1 protects against polyglutamine toxicity induced by other proteins (e.g. mutant Androgen receptor in SBMA models).

Reviewer #4 (Remarks to the Author):

This manuscript entitled “A genome-wide screen in pluripotent cells identifies Mtf1 as a novel suppressor of mutant huntingtin toxicity” uses a clever and fairly stringent survival screen for suppressors of toxicity in a new Q128 Huntington disease mouse ES cell model. The screen itself utilized an MSCV system that upon random integration upregulates neighborhood genes. This revealed possible potent suppressors of HTT N-terminal Q128 toxicity in the presence of other stressors.

The major strength of the work is its use of an unbiased, genome-wide approach followed by additional efforts to validate a top hit in both zebrafish and R6/2 mice, the latter even using AAV/Php.Eb to achieve CNS expression of Mtf1 through venous delivery. This provides a nice proof-of-concept pipeline for a forward genetic screen in culture to efficient *in vivo* validation that others could follow.

One limitation of the study is that it does not examine mutant HTT aggregation (either biochemically or histologically), an important aspect of Huntington disease pathogenesis, and how it might be affected by Mtf1 expression. In addition, there is also only limited evidence showing that protective effects of Mtf1 are specific to mutant HTT and this could be characterized further in the non-mutant HTT settings. The study is otherwise technically sound and their overall conclusions appear to be well supported by their experiments. The manuscript is written well and easy to follow with several helpful diagrams provided along the way. Here are some suggestions that could help address the above limitations along with other minor issues:

1. The study should strongly consider characterizing mutant HTT aggregation phenotypes in these different systems to determine if Mtf1 has effects on this important aspect of HD pathogenesis. Reporting such findings would significantly strengthen the study towards understanding why Mtf1 is protective and if it's acting upstream or downstream of misfolded HTT accumulation.

- a. Figure 1b shows lower Q128 monomeric levels, suggesting there may be increased aggregation in the ES cells. Figure 4c does not demonstrate any change in the monomer levels, so perhaps the effects of these suppressors could be independent of the propensity of mutant HTT to aggregate. Nonetheless, evaluation of the Q128 cell model by filter trap, evaluation of the stack of the western blot, and/or by immunocytochemistry would be worth reporting. ES cells may not show much or any aggregation phenotypes.

We agree on the importance of characterising mutant HTT aggregation phenotypes in our systems. We first looked for mHTT protein aggregates both by evaluation of the stack of the western blot and by immunofluorescence (EM48 antibody, recognising aggregates³⁰). As predicted by the Reviewer, we could not detect any aggregation phenotype (Supplementary Fig. 1b and Reviewer Fig. 3).

We carefully read several publications about pluripotent stem cells used as models of HD^{17,18,24–26} and none reported the formation of aggregates in undifferentiated cells. We then quantified the levels of monomeric HTT in Fig. 2g and 4d and found only a mild reduction of ~15%. Thus, the protective effect in ES cells seems to be independent of mHTT aggregation.

b. Similarly, there appears to be punctate staining in the Q74 condition in zebrafish (Figure 6b) that is reduced by Mtf1. Are these aggregated proteins? A closer examination of this potential phenotype would be helpful.

We might have presented the data not clearly. The signal shown in Figure 6b is acridine orange staining, which marks with bright dots the apoptotic cells. As shown in Supplementary Fig. 7b, embryos injected with Q74eGFP mRNA show a widespread GFP signal with no punctate staining.

Prompted by Reviewer's comment, we inspected several additional images and never observed punctate GFP signal, after injection of up to 500pg Q74eGFP mRNA/embryo (Reviewer Fig. 3). We are aware of previous zebrafish HD models in which formation of aggregates was reported³¹⁻³³. However, in our study we analysed embryos after 24 hours, while other studies reported aggregates at much later time points³³. Furthermore, other studies were based on injections of higher doses of mRNA encoding for mHTT³² or of mRNA containing a longer polyQ (Q102³²) than the one we used. All these differences might explain the lack of aggregate formation in our system.

Reviewer Figure 3

GFP signal in embryos injected with 500pg Q74eGFP mRNA/embryo. We could not observe punctate staining (i.e. potential aggregates) neither in malformed embryos (top) or in healthy ones.

c. Evaluation of the R6/2 mouse brains by western blotting and immunohistochemistry for mutant HTT aggregation and inclusion formation, respectively, would again help examine if Mtf1 affects mutant HTT aggregation. Moreover, it would also further reveal the extent to which this AAV delivery drives Mtf1 expression in neurons versus non-neuronal cell types and distribution (e.g. cortex versus striatum). Finally, IHC would help determine if there is rescue of neurodegeneration phenotypes (e.g by DARPP32 staining of the striatum) beyond only the behavioral phenotypes reported.

As suggested, we analysed the protein aggregates in R6/2 mice and detected robust formation of aggregates by evaluation of the stack of the western blot and by immunofluorescence, both in the striatum and the cortex (Fig. 7h-j and Supplementary Fig. 8e-f). Delivery of Mtf1, which was confirmed both in the striatum and the cortex (Supplementary Fig. 8d) significantly reduced mHTT protein aggregation in the striatum (Fig. 7h-j).

Interestingly, several studies linked protein aggregation and oxidative stress in neurodegenerative diseases^{28,29}. We investigated ROS production in R6/2 mice and observed increased production, relative to wild-type littermates, which was normalised by Mtf1 delivery (Fig. 7f-g), potentially linking oxidative stress and protein aggregation in R6/2 mice.

We then looked at two markers of autophagy, LC3 and ATG7, and found that both were induced by Mtf1 in the striatum of R6/2 mice (Reviewer Fig. 2). However, these results about autophagy are preliminary and we intend to explore further the role of autophagy in this context in the future.

We measured DARPP32 protein levels, which were unaffected by MTF1 (Reviewer Fig. 3). However, DARPP32 is an early marker of HD pathogenesis³⁴ and AAV-Mtf1 might be administered too late to counteract those early effects.

Reviewer Figure 3

Representative western blotting of DARPP-32 protein in striatal tissues from the indicated mice at 8 weeks of age.

We also measured brain weight, as an indirect measure of brain atrophy, which was significantly reduced in R6/2 mice transduced with AAV-GFP, relative to control littermates (Fig. 7e). Crucially, R6/2 mice transduced with AAV-Mtf1 showed only a partial reduction, which did not reach significance, indicating a protective effect of Mtf1.

2. A relatively underexplored control in this study is the effects of the putative suppressors on Q15-ES cells. This is briefly addressed in figure 5e and supplemental figure 5, which show no significant effects of Mtf1 expression on Q15 or the ES parental line. It is somewhat difficult with only these pieces of data alone to confidently know if Mtf1 effects are mutant HTT-specific versus protective against more generalized toxicity, especially given the noisiness of the data in figure 5e.

We agree on the importance of knowing “if Mtf1 effects are mutant HTT-specific versus protective against more generalized toxicity”.

We thus generated Q15-ES cells expressing an empty vector or a vector encoding the mCherry or a vector encoding for Mtf1. After verifying Mtf1 was overexpressed and its direct targets *Mt1* and *Mt2* were strongly induced (Supplementary Fig. 4b), we exposed cells to 7 different compounds inducing cell death. Notably, Mtf1 did not show any protective effect (Supplementary Fig. 4d).

We conclude that in wild-type cells, which do not express mHTT, Mtf1 does not exert any protective effect against cellular stress, while it shows protective effects in several models (mES cells, zebrafish, human NPCs and mouse models) in which mHTT causes cellular stress and death.

We also performed transcriptomic analysis on both Q15 and Q128 cells expressing Mtf1 and failed to detect significant activation of genes involved in proliferation or apoptosis (Supplementary Fig. 5b-c), further indicating that MTF1 does confer a generic resistance to cell death.

Finally, we expressed Mtf1 in human NPCs and analysed the levels of ROS (DCFDA) and apoptosis (Annexin V) and failed to observe significant effects (Reviewer Fig. 4), further indicating that Mtf1 does not exert a generic protective effect.

Reviewer Figure 4

a, Measurement of ROS production via DCFDA staining in NPC Q21_EV and NPC Q21_Mtf1. Representative flow cytometry profiles are represented in the right panels. Bars indicate the mean of 4 biological replicates. For each biological replicate, technical replicates are shown as dots of different colours.

b, Measurement of cell death by Annexin V uptake and Flow Cytometry. Fold-changes were calculated relative to the NPC Q21_EV samples. Representative flow cytometry profiles are on the right. Bars indicate the mean of 3 independent experiments shown as dots (technical replicates of different experiments are presented with different colour). Fold-changes were calculated relative to the NPC Q21_EV samples. P-values were calculated with Two-tailed unpaired T-test.

a. In figure 4e, can Mtf1 also rescue toxicity in Q15_EV with MG132/tamoxifen? Or is this effect only seen in Q128? This can be similarly asked for another putative suppressor. If so, this might suggest that Mtf1 is generally protective against general stress (as alluded to by the authors in discussion).

We performed the required experiments, in which Q15 cells were exposed to the stressors MG132 and Tamoxifen (Supplementary Fig. 4d) and observed no protective effect of Mtf1.

In contrast, MTF1 rescues specifically cellular stress induced by mHTT (Fig. 4e-f).

b. Consider moving both Figure 5e and supplemental figure 5 to figure 4, where one would first ask if the effects of the suppressors are Q128-specific.

Thanks for the good suggestion, we reshuffled the figures in order to discuss first the specificity of Mtf1 effects (Fig. 4h and Supplementary Fig. 4c).

3. There appears to be a robust effect on Cresyl violet intensity with Kdm5b in figure 4e, but no significant effect on the number of cells in Figure 4d - why? Perhaps reporting the results of n=3 experiments in 4e could help resolve the discrepancy (rather than showing a single experiment with technical duplicate in 4e). Relatedly, the comparisons in figure 4e are difficult to follow. Consider breaking apart Figure 4e with separate panels for untreated, MG132, and Tamoxifen treated conditions, so Q15, Q128, and Q128_Mtf1 etc can be compared directly side-by-side.

We agree on the difficulty of following all comparisons in Figures 4d and 4e, therefore we have divided them into separate figures. Also, as suggested, now we show at least 3 independent experiments. So the new Fig. 4e-f and Supplementary Fig. 4a shows the effect of candidates without stressors, while Figure 4g shows the effects of candidates in the presence of either Tamoxifen or MG132.

Concerning Kdm5b, our results seem to indicate that it does not confer significant protection against mHTT, but it protects against stressors by themselves, which explains why it was found in the screening, but it is not worth following up.

4. This manuscript indicates Mtf1 is a transcription factor that translocates to the nucleus during stress - can this be assessed by immunocytochemistry in the Q128 versus Q15 condition? Also can the authors comment on Mtf1 conservations across species, especially since murine Mtf1 appears to have an effect in zebrafish? Is there an ortholog in zebrafish? This could be included in the discussion.

We assessed the localization of MTF1 in both Q15 and Q128 cells and found that it is mostly nuclear in both cell lines, with Q128 cells showing slightly higher nuclear signal (Reviewer Fig. 1). This is in line with Cd accumulation and basal level of ROS production, which are both higher in Q128 cells. We should also bear in mind that these are overexpression experiments, thus the physiological regulation of Mtf1 might be

altered by the high levels of its expression. We will investigate in more detail how mHTT regulates MTF1 localisation.

Concerning the evolutionary conservation of Mtf1, the first MTF1 cDNA was cloned and characterised in mouse in 1993³⁵. In the subsequent years MTF1 was identified and characterised in many other vertebrate and invertebrate organisms like human³⁶, fish^{15,37,38}, insects³⁹ and molluscs^{40,41}. MTF1 is broadly characterised by an evolutionary conserved DNA binding domain composed of six zinc fingers of Cys2His2-type, a transcriptional activation domain⁴². The zinc finger domain is conserved in all species analysed so far and it was demonstrated that mammalian and *Drosophila* MTF1 can cross-complement each other when tested in the respective knock-out background⁴³. Similarly, expression of MTF1 from pufferfish *Takifugu rubripes* in MTF-1 null mutant mouse cells induced the transcription of a mouse MT1 promoter, an effect boosted by zinc and cadmium induction¹⁵.

We expressed mouse *Mtf1* mRNA in murine, human and zebrafish models, and observed consistent protective effects and activation of target genes, such as MTs. Mouse MTF1 shows high identity of 92, 93 and 99% in the zinc finger domain with fugu, zebrafish and human (Supplementary Fig. 7a). We conclude the high evolutionary conservation of zinc finger DNA binding domain of MTF1 confers the capacity to activate target genes when MTF1 is expressed in different species, which we and others observed^{15,43}.

5. In figure 5, the authors show rescue of mutant HTT-induced transcriptional changes by Mtf1 overexpression. However, what are the total number of DEGs in (Q128_MTF1 vs Q15_EV) and (Q128_EV vs Q15_EV)?

Is the rescue of DEGs in Q128_Mtf1 also associated with fewer total DEGs? Or does Mtf1 overexpression drive a separate robust transcriptional program with even more DEGs? A small supplementary bar graph of the total DEGs in these conditions could show this.

We calculated the number of DEGs and showed them in Supplementary Fig. 5a.

As correctly predicted by the Reviewer, MTF1 drives a separate robust transcriptional program with more DEGs than those controlled by mHTT.

b. Similarly, are RNA-seq data available for Mtf1 expression by itself or in the Q15 condition? Would that lead to a similar gene enrichment analysis in figure 5d? It would be helpful to know if there is any overlap with the DEGs of the Q128 line (relative to Q15), especially if DEGs are moving in the opposing direction.

To further test whether Mtf1 exerts a general protective effect, regardless of mHTT, we generated Q15 cells expressing Mtf1, or an Empty vector, and performed transcriptome analysis. We identified genes regulated by Mtf1 in Q15 cells and in Q128 cells and observed a limited overlap (Supplementary Fig. 5d).

Among 1571 genes were differentially regulated ($FC > |0.5|$ and $p\text{-value} < 0.05$) in either Q15 or Q128 cells, the fraction of those regulated in both cell lines was significantly underrepresented (395 genes, $p\text{-value} = 7.1e-53$ Binomial test, Supplementary Fig.

5d). Of note, only 13 genes moved in the opposite direction, indicating that Mtf1 very rarely switches from activation to repression of the same target gene.

We performed a more comprehensive analysis of processes regulated by Mtf1 in Q128 (Supplementary Fig. 5b). We added a column indicating whether the same processes were also regulated by Mtf1 in Q15 cells. Only some processes (e.g. metal and steroids homeostasis) were regulated by Mtf1 also in Q15 cells. Interestingly, genes associated with glutathione metabolism were regulated by Mtf1 only in Q128 cells.

We conclude that Mtf1 controls only partially overlapping processes in Q15 and Q128 cells. Among those controlled in both cell lines we could not find processes linked to proliferation or apoptosis (Supplementary Fig. 5b-c), thus ruling out a general protective mechanism exerted by Mtf1.

Other minor comments:

1. Considering moving supplemental figure 4 into the main figure 4 (alongside 4c) if space allows.

We followed the good suggestion. Candidates do not change significantly *mHTT* mRNA levels, and cause a mild reduction in mHTT protein.

2. Consider adding detail in the abstract for the type of model used for the screen. It currently only mentions "pluripotent stem cells," which could be misinterpreted as patient-derived iPSCs.

We clarified that we used murine embryonic stem cells (ESCs) and human NPCs.

4. In Figure 7c,d,e, consider using red lines for both R6/2 mouse conditions so WT AAV-Mtf1 and R6/2 AAV-GFP black dashed lines do not get confused with each other.

We changed the colour of lines, so they do not get confused.

5. For figure 7c,d,e top-half, consider an ANOVA statistical analysis for the longitudinal behavioral study to highlight significant differences.

We added the *p*-values calculated with the suggested statistical test (Fig. 7b-c and Supplementary Fig. 8c). The new tests confirmed that MTF1 significantly counteracted the detrimental effects of mHTT.

6. The *p* value is quite low for Figure 4d, Q128_Fbxo34, despite nearly overlapping error bars - can the authors confirm this is the correct *p* value?

In Figure 4d of the submitted manuscript, we calculated the *p*-value with ANOVA on repeat measures (as suggested by the reviewer for the *in vivo* experiment in the point

5, above). The p value is indeed quite low, but the biological effect is mild, mostly likely because the test detects very small differences.

Now, for clarity, we showed first the effects of each candidate after 96 hours (New Fig. 4e). In this case, the p value calculated with a Student's t test is 0.19 for Q128_Fbx034 vs Q128_EV. Fbxo34 also failed to rescue the mHTT effect in the presence of stressors (Fig. 4g).

7. Figure 3c, it is not clear to me NGS nodes are “pink” as described and would consider using a different color. The HD interacting gene nodes look more pink.

We changed the colours of NGS nodes to green in New Figure 3b, we agree it was a bit confusing.

8. Page 2, line 17 italicize in vitro

We italicised all '*in vivo*' and '*in vitro*'.

9. Page 3, line 26: Define polyQ as polyglutamine for first use. In line 29, can then say polyQ-encoding.

Done

10. Page 3, line 39-41 “For this reason.. reduce mutant toxicity”: separate into two sentences.

Done

11. Page 7, line 109 - “...44 mutant clonal lines *that* emerged...”

Done

REFERENCES

1. An, M. C. *et al.* Genetic Correction of Huntington's Disease Phenotypes in Induced Pluripotent Stem Cells. *Cell Stem Cell* **11**, 253–263 (2012).
2. Xu, X. *et al.* Reversal of Phenotypic Abnormalities by CRISPR/Cas9-Mediated Gene Correction in Huntington Disease Patient-Derived Induced Pluripotent Stem Cells. *Stem Cell Rep.* **8**, 619–633 (2017).
3. Ooi, J. *et al.* Unbiased Profiling of Isogenic Huntington Disease hPSC-Derived CNS and Peripheral Cells Reveals Strong Cell-Type Specificity of CAG Length Effects. *Cell Rep.* **26**, 2494-2508.e7 (2019).
4. Seong, I. S. *et al.* HD CAG repeat implicates a dominant property of huntingtin in mitochondrial energy metabolism. *Hum. Mol. Genet.* **14**, 2871–2880 (2005).
5. Xu, X. *et al.* Reversal of Phenotypic Abnormalities by CRISPR/Cas9-Mediated Gene Correction in Huntington Disease Patient-Derived Induced Pluripotent Stem Cells. *Stem Cell Rep.* **8**, 619–633 (2017).
6. Mason, R. P. *et al.* Glutathione peroxidase activity is neuroprotective in models of Huntington's disease. *Nat. Genet.* **45**, 1249–1254 (2013).
7. Saini, N., Georgiev, O. & Schaffner, W. The parkin mutant phenotype in the fly is largely rescued by metal-responsive transcription factor (MTF-1). *Mol. Cell. Biol.* **31**, 2151–2161 (2011).
8. Bi, Y., Lin, G. X., Millecchia, L. & Ma, Q. Superinduction of metallothionein I by inhibition of protein synthesis: role of a labile repressor in MTF-1 mediated gene transcription. *J. Biochem. Mol. Toxicol.* **20**, 57–68 (2006).
9. Kwakye, G. F. *et al.* Heterozygous huntingtin promotes cadmium neurotoxicity and neurodegeneration in striatal cells via altered metal transport and protein kinase C delta dependent oxidative stress and apoptosis signaling mechanisms. *Neurotoxicology* **70**, 48–61 (2019).
10. Klaassen, C. D., Liu, J. & Diwan, B. A. Metallothionein Protection of Cadmium Toxicity.

- Toxicol. Appl. Pharmacol.* **238**, 215–220 (2009).
11. Wang, B. & Du, Y. Cadmium and its neurotoxic effects. *Oxid. Med. Cell. Longev.* **2013**, 898034 (2013).
 12. Carbognin, E., Betto, R. M., Soriano, M. E., Smith, A. G. & Martello, G. Stat3 promotes mitochondrial transcription and oxidative respiration during maintenance and induction of naive pluripotency. *EMBO J.* **35**, 618–634 (2016).
 13. Davis, B. N., Hilyard, A. C., Lagna, G. & Hata, A. SMAD proteins control DROSHA-mediated microRNA maturation. *Nature* **454**, 56–61 (2008).
 14. Suzuki, H. I. *et al.* Modulation of microRNA processing by p53. *Nature* **460**, 529–533 (2009).
 15. der Maur, A. A. *et al.* Characterization of the mouse gene for the heavy metal-responsive transcription factor MTF-1. *Cell Stress Chaperones* **5**, 196–206 (2000).
 16. Francis, M. & Grider, A. Bioinformatic analysis of the metal response element and zinc-dependent gene regulation via the metal response element-binding transcription factor 1 in Caco-2 cells. *Biometals Int. J. Role Met. Ions Biol. Biochem. Med.* **31**, 639–646 (2018).
 17. Jacobsen, J. C. *et al.* HD CAG-correlated gene expression changes support a simple dominant gain of function. *Hum. Mol. Genet.* **20**, 2846–2860 (2011).
 18. White, J. K. *et al.* Huntingtin is required for neurogenesis and is not impaired by the Huntington's disease CAG expansion. *Nat. Genet.* **17**, 404–410 (1997).
 19. Trettel, F. *et al.* Dominant phenotypes produced by the HD mutation in STHdh(Q111) striatal cells. *Hum. Mol. Genet.* **9**, 2799–2809 (2000).
 20. Williams, B. B. *et al.* Disease-toxicant screen reveals a neuroprotective interaction between Huntington's disease and manganese exposure. *J. Neurochem.* **112**, 227–237 (2010).
 21. Pouladi, M. A. *et al.* Full-length huntingtin levels modulate body weight by influencing insulin-like growth factor 1 expression. *Hum. Mol. Genet.* **19**, 1528–1538 (2010).
 22. Aiken, C. T. *et al.* Phosphorylation of Threonine 3. *J. Biol. Chem.* **284**, 29427–29436

- (2009).
23. Persichetti, F. *et al.* Differential expression of normal and mutant Huntington's disease gene alleles. *Neurobiol. Dis.* **3**, 183–190 (1996).
 24. Ismailoglu, I. *et al.* Huntingtin Protein is Essential for Mitochondrial Metabolism, Bioenergetics and Structure in Murine Embryonic Stem Cells. *Dev. Biol.* **391**, 230–240 (2014).
 25. Biagioli, M. *et al.* Htt CAG repeat expansion confers pleiotropic gains of mutant huntingtin function in chromatin regulation. *Hum. Mol. Genet.* **24**, 2442–2457 (2015).
 26. Pryor, W. M. *et al.* Huntingtin promotes mTORC1 signaling in the pathogenesis of Huntington's disease. *Sci. Signal.* **7**, ra103 (2014).
 27. Lu, B. & Palacino, J. A novel human embryonic stem cell-derived Huntington's disease neuronal model exhibits mutant huntingtin (mHTT) aggregates and soluble mHTT-dependent neurodegeneration. *FASEB J.* **27**, 1820–1829 (2013).
 28. Lévy, E. *et al.* Causative Links between Protein Aggregation and Oxidative Stress: A Review. *Int. J. Mol. Sci.* **20**, E3896 (2019).
 29. Guerrero-Gómez, D. *et al.* Loss of glutathione redox homeostasis impairs proteostasis by inhibiting autophagy-dependent protein degradation. *Cell Death Differ.* **26**, 1545–1565 (2019).
 30. Gutekunst, C. A. *et al.* Nuclear and neuropil aggregates in Huntington's disease: relationship to neuropathology. *J. Neurosci. Off. J. Soc. Neurosci.* **19**, 2522–2534 (1999).
 31. Miller, V. M. *et al.* CHIP Suppresses Polyglutamine Aggregation and Toxicity In Vitro and In Vivo. *J. Neurosci.* **25**, 9152–9161 (2005).
 32. Schiffer, N. W. *et al.* Identification of Anti-prion Compounds as Efficient Inhibitors of Polyglutamine Protein Aggregation in a Zebrafish Model*. *J. Biol. Chem.* **282**, 9195–9203 (2007).
 33. Veldman, M. B. *et al.* The N17 domain mitigates nuclear toxicity in a novel zebrafish Huntington's disease model. *Mol. Neurodegener.* **10**, 67 (2015).

34. Bibb, J. A. *et al.* Severe deficiencies in dopamine signaling in presymptomatic Huntington's disease mice. *Proc. Natl. Acad. Sci.* **97**, 6809–6814 (2000).
35. Radtke, F. *et al.* Cloned transcription factor MTF-1 activates the mouse metallothionein I promoter. *EMBO J.* **12**, 1355–1362 (1993).
36. Brugnera, E. *et al.* Cloning, chromosomal mapping and characterization of the human metal-regulatory transcription factor MTF-1. *Nucleic Acids Res.* **22**, 3167–3173 (1994).
37. Chen, W.-Y., John, J. A. C., Lin, C.-H. & Chang, C.-Y. Molecular cloning and developmental expression of zinc finger transcription factor MTF-1 gene in zebrafish, *Danio rerio*. *Biochem. Biophys. Res. Commun.* **291**, 798–805 (2002).
38. Cheung, A. P.-L., Au, C. Y.-M., Chan, W. W.-L. & Chan, K. M. Characterization and localization of metal-responsive-element-binding transcription factors from tilapia. *Aquat. Toxicol. Amst. Neth.* **99**, 42–55 (2010).
39. Zhang, B., Egli, D., Georgiev, O. & Schaffner, W. The Drosophila homolog of mammalian zinc finger factor MTF-1 activates transcription in response to heavy metals. *Mol. Cell. Biol.* **21**, 4505–4514 (2001).
40. Lee, S. Y. & Nam, Y. K. Molecular cloning of metal-responsive transcription factor-1 (MTF-1) and transcriptional responses to metal and heat stresses in Pacific abalone, *Haliotis discus hannai*. *Fish. Aquat. Sci.* **20**, 9 (2017).
41. Meng, J. *et al.* Transcription factor CgMTF-1 regulates CgZnT1 and CgMT expression in Pacific oyster (*Crassostrea gigas*) under zinc stress. *Aquat. Toxicol. Amst. Neth.* **165**, 179–188 (2015).
42. Günther, V., Lindert, U. & Schaffner, W. The taste of heavy metals: gene regulation by MTF-1. *Biochim. Biophys. Acta* **1823**, 1416–1425 (2012).
43. Balamurugan, K. *et al.* Metal-responsive transcription factor (MTF-1) and heavy metal stress response in Drosophila and mammalian cells: a functional comparison. *Biol. Chem.* **385**, 597–603 (2004).

REVIEWERS' COMMENTS

Reviewer #1 (Remarks to the Author):

The authors have addressed my biggest concern, the lack of comparison with a human HD model without fragment expression. They have added data using 21Q or 109Q iPSCs differentiated to NPCs. The ROS production results are ok (Fig. 8e), the cell death ones (Fig. 8f) a bit disappointing. In any case, it is still far fetched to conclude much from a marginally increased cell death in NPCs about the degeneration of cell in the striatum at a much later time point.

I found the following aspects confusing about Fig 8:

- Fig 8d: This figure is almost the same figure as shown in Suppl Fig. 9c. The figure legend seems identical, but the data is slightly different (and so is the p-value).
- Fig 8c: The Nestin stains (and qPCRs) are confusing. Nestin seems to be expressed in iPSCs, or there is nonspecific staining, but it doesn't look very different to the NPC case. In Suppl. Fig. 9a, Nestin is only significantly upregulated for the Q21 line.
- Fig 8: Panel f should come before panel g.

Reviewer #2 (Remarks to the Author):

I think the authors addressed most of my comments in a satisfactory manner: the quality of the work substantially improved. I believe the manuscript is now ready for acceptance in Nature Communications

Reviewer #3 (Remarks to the Author):

The authors have responded to the reviewers' comments in a suitable and satisfactory way.

Reviewer #4 (Remarks to the Author):

The authors adequately and appropriately responded to my initial concerns with detailed and thoughtful explanations. The additional experiments significantly strengthen the claims of the manuscript. The evidence is quite compelling that MTF1 overexpression is protective in HD, although the reasons for this still aren't exactly clear and may be related to homeostasis of cellular metals and ROS.

Just a few other minor points when going through this updated manuscript:

1. In the figure 7j, the immunofluorescence data of the R6/2 mouse experiments are not really clear. What is the staining by EM48 being shown? Is it total levels and is there any staining in WT mice? (This could be shown in supplemental). Also a closer look at the nuclei could help establish whether there is just total reduced staining or if there is a reduction in the size and frequency of nuclear inclusions, which should be abundant in the striatum. Also I assume that the inset is of DAPI, but that does not provide any information to the localization of the red fluorescent signal.
2. Results,, line 338 and figure 7e- It is a little confusing to state that there is indication of a protect effect of Mtf1 while saying it did not reach statistical significance. Can the authors in some way show that there is a statistically significant rescue in R6/2 Mtf1 relative to R6/2 GFP?

3. Results line 345 - "*As* (or *since*) MTF1 lowers ROS production in Q128 cells..."

4. Line 406 - "MTF1 translocateS"

POINT-BY-POINT RESPONSE TO REVIEWERS' COMMENTS

Reviewer #1 (Remarks to the Author):

The authors have addressed my biggest concern, the lack of comparison with a human HD model without fragment expression. They have added data using 21Q or 109Q iPSCs differentiated to NPCs. The ROS production results are ok (Fig. 8e), the cell death ones (Fig. 8f) a bit disappointing. In any case, it is still far fetched to conclude much from a marginally increased cell death in NPCs about the degeneration of cell in the striatum at a much later time point.

I found the following aspects confusing about Fig 8:

- Fig 8d: This figure is almost the same figure as shown in Suppl Fig. 9c. The figure legend seems identical, but the data is slightly different (and so is the p-value).

Figure 8d describes the proliferation of iPSCs, while Figure 9c the proliferation of NPCs. This is clearly indicated in both figure panels. We edited the text to avoid any misunderstanding. See Page 17.

- Fig 8c: The Nestin stains (and qPCRs) are confusing. Nestin seems to be expressed in iPSCs, or there is nonspecific staining, but it doesn't look very different to the NPC case. In Suppl. Fig. 9a, Nestin is only significantly upregulated for the Q21 line.

Nestin is indeed expressed both in iPSCs and NPCs. We confirmed this by qPCR and Immunostaining and also by RNAseq (data not shown).

Nestin is widely used as a marker of (cancer) stem cells (Neradil et al., 2015, Michalczyk et al., 2005). Indeed other publications (e.g. Ooi et al. 2019) claim the presence of Nestin in NPCs, while we could not find publications claiming the absence of Nestin in iPSCs.

The co-expression of Nestin and SOX1 is commonly used to identify NPCs.

- Fig 8: Panel f should come before panel g.

We made the changes suggested by the Reviewer.

Reviewer #2 (Remarks to the Author):

I think the authors addressed most of my comments in a satisfactory manner: the quality of the work substantially improved. I believe the manuscript is now ready for acceptance in Nature Communications

Reviewer #3 (Remarks to the Author):

The authors have responded to the reviewers' comments in a suitable and satisfactory way.

Reviewer #4 (Remarks to the Author):

The authors adequately and appropriately responded to my initial concerns with detailed and thoughtful explanations. The additional experiments significantly strengthen the claims of the manuscript. The evidence is quite compelling that MTF1 overexpression is protective in HD, although the reasons for this still aren't exactly clear and may be related to homeostasis of cellular metals and ROS.

Just a few other minor points when going through this updated manuscript:

1. In the figure 7j, the immunofluorescence data of the R6/2 mouse experiments are not really clear.

What is the staining by EM48 being shown?

Is it total levels and is there any staining in WT mice? (This could be shown in supplemental).

Also a closer look at the nuclei could help establish whether there is just total reduced staining or if there is a reduction in the size and frequency of nuclear inclusions, which should be abundant in the striatum.

Also I assume that the inset is of DAPI, but that does not provide any information to the localization of the red fluorescent signal.

The EM48 antibody recognises the insoluble fraction of HTT that is found in aggregates. EM48 does not recognise total HTT. We changed the panel to make this clear.

Following Reviewer's suggestion, now we show EM48, the nuclei marked by DAPI, and the two signals merged together. It is now clear that mutant HTT aggregates are mostly nuclear, as expected in the striatum.

We also included WT mice, which are devoid of HTT aggregates, as expected.

We quantified the frequency of nuclear inclusions (i.e. number of aggregates for each nucleus) and observed a significant decrease. Please see New Figure 7h-i.

2. Results, line 338 and figure 7e- It is a little confusing to state that there is indication of a protect effect of Mtf1 while saying it did not reach statistical significance. Can the authors in some way show that there is a statistically significant rescue in R6/2 Mtf1 relative to R6/2 GFP?

We apologise for the confusion.

We defined a protective effect of Mtf1 when experimental measurements relative to R6/2 mice treated with AAV-Mtf1 were not significantly different from WT.

However if we consider the comparison between R6/2 AAV-GFP and R6/2 AAV-Mtf1 separately, as implied by the reviewer, we obtain a p-value of 0.019 (two tailed unpaired T-test), indicating a protective effect of Mtf1. We edited the text accordingly (see pag 16).

3. Results line 345 - "**As* (or *since*) MTF1 lowers ROS production in Q128 cells..."

We made the changes suggested by the Reviewer.

4. Line 406 - "MTF1 translocateS"

We made the changes suggested by the Reviewer.